# Learning stackable and skippable LEGO bricks for efficient, reconfigurable, and variable-resolution diffusion modeling

**Huangjie Zheng**[1,2]**, Zhendong Wang**[1]**, Jianbo Yuan**[2]**, Guanghan Ning**[2]
**Pengcheng He**[3]**, Quanzeng You**[2]**, Hongxia Yang**[2]**, Mingyuan Zhou**[1]
[1]The University of Texas at Austin, [2]ByteDance Inc., [3]Microsoft Azure AI
`{huangjie.zheng,zhendong.wang}@utexas.edu`
`{jianbo.yuan,guanghan.ning,quanzeng.you,hx.yang}@bytedance.com`
`herbert.he@gmail.com, mingyuan.zhou@mccombs.utexas.edu`

## Abstract

Diffusion models excel at generating photo-realistic images but come with significant computational costs in both training and sampling. While various techniques address these computational challenges, a less-explored issue is designing an efficient and adaptable network backbone for iterative refinement. Current options like U-Net and Vision Transformer often rely on resource-intensive deep networks and lack the flexibility needed for generating images at variable resolutions or with a smaller network than used in training. This study introduces LEGO bricks, which seamlessly integrate Local-feature Enrichment and Global-content Orchestration. These bricks can be stacked to create a test-time reconfigurable diffusion backbone, allowing selective skipping of bricks to reduce sampling costs and generate higher-resolution images than the training data. LEGO bricks enrich local regions with an MLP and transform them using a Transformer block while maintaining a consistent full-resolution image across all bricks. Experimental results demonstrate that LEGO bricks enhance training efficiency, expedite convergence, and facilitate variable-resolution image generation while maintaining strong generative performance. Moreover, LEGO significantly reduces sampling time compared to other methods, establishing it as a valuable enhancement for diffusion models. Our code and project page are available at `https://jegzheng.github.io/LEGODiffusion`.

## 1 Introduction

Diffusion models, also known as score-based generative models, have gained significant traction in various domains thanks to their proven effectiveness in generating high-dimensional data and offering simple implementation (Sohl-Dickstein et al., 2015; Song & Ermon, 2019; 2020; Ho et al., 2020; Song et al., 2021). Decomposing data generation into simple denoising tasks across evolving timesteps with varying noise levels, they have played a pivotal role in driving progress in a wide range of fields, with a particular emphasis on their contributions to image generation (Dhariwal & Nichol, 2021; Nichol et al., 2021; Ramesh et al., 2022; Saharia et al., 2022; Rombach et al., 2022).

Despite these advancements, diffusion models continue to face a significant challenge: their substantial computational requirements, not only during training but also during sampling. In sampling, diffusion models typically demand a large number of functional evaluations (NFE) to simulate the reverse diffusion process. To tackle this challenge, various strategies have been proposed, including methods aimed at expediting sampling through approximated reverse steps (Song et al., 2020; Zhang & Chen, 2022; Lu et al., 2022; Karras et al., 2022), as well as approaches that integrate diffusion with other generative models (Zheng et al., 2022c; Pandey et al., 2022).

Another clear limitation is that the network used in sampling must generally be the same as that used in training, unless progressive distillation techniques are applied (Salimans & Ho, 2022; Song et al., 2023). With or without distillation, the network used for sampling typically lacks reconfigurability and would require retraining if a smaller memory or computational footprint is desired.

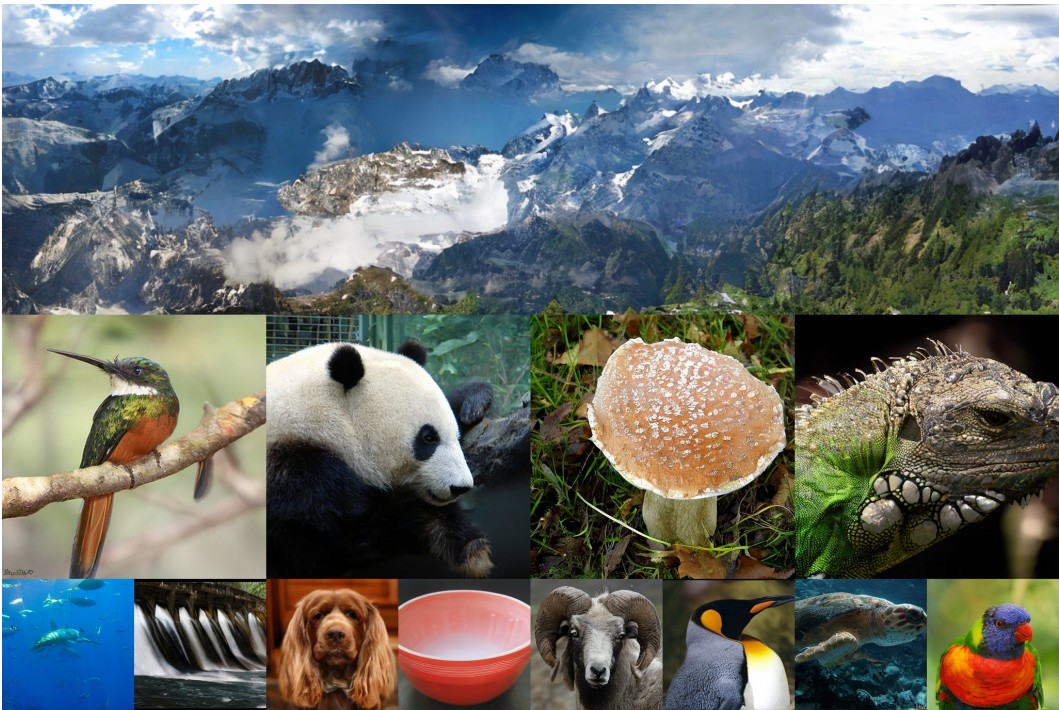

Figure 1: Visualization of LEGO-generated images. Top Row: $2048 \times 600$ panorama image sample, from the model trained on ImageNet $256 \times 256$. Middle Row: $512 \times 512$ image samples, trained on ImageNet $512 \times 512$. Bottom Row: $256 \times 256$ image samples, trained on ImageNet $256 \times 256$.

Beside the computing-intensive and rigid sampling process, training diffusion models also entails a considerable number of iterations to attain convergence to an acceptable checkpoint. This requirement arises from the model's need to learn how to predict the mean of clean images conditioned on noisy inputs, which exhibit varying levels of noise. This concept can be concluded from either the Tweedie's formula (Robbins, 1992; Efron, 2011), as discussed in Luo (2022) and Chung et al. (2022), or the Bregman divergence (Banerjee et al., 2005), as illustrated in Zhou et al. (2023). This mechanism imposes a significant computational burden, particularly when dealing with larger and higher dimensional datasets. To mitigate the training cost, commonly-used approaches include training the diffusion model in a lower-dimensional space and then recovering the high-dimensional data using techniques such as pre-trained auto-encoders (Rombach et al., 2022; Li et al., 2022b) or cascaded super-resolution models (Ho et al., 2022; Saharia et al., 2022). An alternative strategy involves training diffusion models at the patch-level input (Luhman & Luhman, 2022), requiring additional methods to ensure generated samples maintain a coherent global structure. These methods may include incorporating full-size images periodically (Wang et al., 2023), using encoded or downsampled semantic features (Ding et al., 2023; Arakawa et al., 2023), or employing masked transformer prediction (Zheng et al., 2023; Gao et al., 2023).

Despite numerous efforts to expedite diffusion models, a significant but relatively less-explored challenge remains in designing an efficient and flexible network backbone for iterative refinement during both training and sampling. Historically, the dominant choice for a backbone has been based on U-Net (Ronneberger et al., 2015), but recently, an alternative backbone based on the Vision Transformer (ViT) (Dosovitskiy et al., 2021) has emerged as a compelling option. However, it's important to note that both U-Net and ViT models still require the inclusion of deep networks with a substantial number of computation-intensive convolutional layers or Transformer blocks to achieve satisfactory performance. Moreover, it's important to highlight that neither of these options readily allows for network reconfiguration during sampling, and generating images at higher resolutions than the training images frequently poses a significant challenge.

Our primary objective is to introduce the "LEGO brick," a fundamental network unit that seamlessly integrates two essential components: **L**ocal-feature **E**nrichment and **G**lobal-content **O**rchestration. These bricks can be vertically stacked to form the reconfigurable backbone of a diffusion model that introduces spatial refinement within each timestep. This versatile backbone not only enables selective

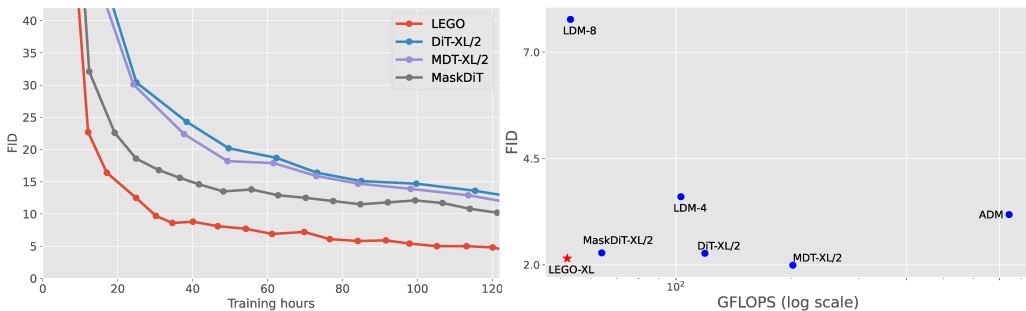

Figure 2: With classifier-free guidance, we present the following: Left Panel: A comparison of convergence, measured with FID versus training time. Right Panel: A comparison of the computation cost measured with FID versus training FLOPs. These experiments were conducted using eight NVIDIA A100 GPUs on ImageNet ($256 \times 256$), with a batch size of 256.

skipping of LEGO bricks for reduced sampling costs but also capably generates images at resolutions significantly higher than the training set.

A LEGO diffusion model is assembled by stacking a series of LEGO bricks, each with different input sizes. Each brick processes local patches whose dimensions are determined by its individual input size, all aimed at refining the spatial information of the full-resolution image. In the process of "Local-feature Enrichment," a specific LEGO brick takes a patch that matches its input size ($e.g.$, a $16 \times 16$ noisy patch) along with a prediction from the preceding brick. This patch-level input is divided into non-overlapping local receptive fields ($e.g.$, four $8 \times 8$ patches), and these are represented using local feature vectors projected with a "token" embedding layer. Following that, "Global-content Orchestration" involves using Transformer blocks (Vaswani et al., 2017; Dosovitskiy et al., 2021), comprising multi-head attention and MLP layers, to re-aggregate these local token-embedding feature vectors into a spatially refined output that matches the input size. This approach results in an efficient network unit achieved through MLP mixing that emphasizes local regions while maintaining a short attention span ($e.g.$, a sequence of four token-embedding vectors). Importantly, each LEGO brick is trained using sampled input patches rather than entire images, significantly reducing the computational cost associated with the model's forward pass. This approach not only enables the flexible utilization of LEGO bricks of varying sizes during training but also facilitates a unique reconfigurable architecture during generation. Furthermore, it empowers the model with the capability to generate images at significantly higher resolutions than those present in the training dataset, as illustrated in Figure 1.

In summary, the stackable and skippable LEGO bricks possess several noteworthy characteristics in managing computational costs in both training and sampling, offering a flexible and test-time reconfigurable backbone, and facilitating variable-resolution generation. Our experimental evaluations, as shown in Figures 1 and 2, clearly demonstrate that LEGO strikes a good balance between efficiency and performance across challenging image benchmarks. LEGO significantly enhances training efficiency, as evidenced by reduced FLOPs, faster convergence, and shorter training times, all while maintaining a robust generation performance. These advantages extend seamlessly into the sampling phase, where LEGO achieves a noteworthy 60% reduction in sampling time compared to DiT (Peebles & Xie, 2022), while keeping the same NFE. Additionally, LEGO has the capability to generate images at much higher resolutions ($e.g.$, $2048 \times 600$) than training images ($e.g.$, $256 \times 256$).

## 2 LOCAL-FEATURE ENRICHMENT AND GLOBAL-CONTENT ORCHESTRATION

### 2.1 MOTIVATIONS FOR CONSTRUCTING LEGO BRICKS

Both U-Net and ViT, which are two prominent architectural choices for diffusion models, impose a significant computational burden. The primary goal of this paper is to craft an efficient diffusion-modeling architecture that seamlessly integrates the principles of local feature enrichment, progressive spatial refinement, and iterative denoising within a unified framework. Additionally, our objective extends to a noteworthy capability: the selective skipping of LEGO bricks, which operate on different patch sizes at various time steps during the generation process. This unique test-time reconfigurable capability sets our design apart from prior approaches.

Our envisioned LEGO bricks are intended to possess several advantageous properties: **1) Spatial Efficiency in Training:** Within the ensemble, the majority of LEGO bricks are dedicated to producing local patches using computation-light MLP mixing and attention modules. This design choice leads to a significant reduction in computational Floating-Point Operations (FLOPs) and substantially shortens the overall training duration. **2) Spatial Efficiency in Sampling:** During sampling, the LEGO bricks can be selectively skipped at each time step without a discernible decline in generation performance. Specifically, when $t$ is large, indicating greater uncertainty in the global spatial structure, more patch-level LEGO bricks can be safely skipped. Conversely, when $t$ is small, signifying a more stable global spatial structure, more full-resolution LEGO bricks can be bypassed. **3) Versatility:** LEGO bricks showcase remarkable versatility, accommodating both end-to-end training and sequential training from lower to upper bricks, all while enabling generation at resolutions significantly higher than those employed during training. Furthermore, they readily support the integration of existing pre-trained models as LEGO bricks, enhancing the model's adaptability and ease of use.

## 2.2 Technical Preliminary of Diffusion-based Generative Models

Diffusion-based generative models employ a forward diffusion chain to gradually corrupt the data into noise, and an iterative-refinement-based reverse diffusion chain to regenerate the data. We use $\alpha_t \in [0, 1]$, a decreasing sequence, to define the variance schedule $\{\beta_t = 1 - \frac{\alpha_t}{\alpha_{t-1}}\}_{t=1}^T$ and denote $\mathbf{x}_0$ as a clean image and $\mathbf{x}_t$ as a noisy image. The reverse diffusion chain can be optimized by maximizing the evidence lower bound (ELBO) (Blei et al., 2017) of a variational autoencoder (Kingma & Welling, 2014), using a hierarchical prior with $T$ stochastic layers (Ho et al., 2020; Song et al., 2020; 2021; Kingma et al., 2021; Zhou et al., 2023). Let's represent the data distribution as $q(\mathbf{x}_0)$ and the generative prior as $p(\mathbf{x}_T)$. The forward and reverse diffusion processes discretized into $T$ steps can be expressed as:

$$\text{Forward}: q(\mathbf{x}_{0:T}) = q(\mathbf{x}_0) \prod_{t=1}^T q(\mathbf{x}_t \mid \mathbf{x}_{t-1}) = q(\mathbf{x}_0) \prod_{t=1}^T \mathcal{N}\left(\mathbf{x}_t; \frac{\sqrt{\alpha_t}}{\sqrt{\alpha_{t-1}}}\mathbf{x}_{t-1}, 1 - \frac{\alpha_t}{\alpha_{t-1}}\right), \quad (1)$$

$$\text{Reverse}: p_\theta(\mathbf{x}_{0:T}) = p(\mathbf{x}_T) \prod_{t=1}^T p_\theta(\mathbf{x}_{t-1} \mid \mathbf{x}_t) = p(\mathbf{x}_T) \prod_{t=1}^T q(\mathbf{x}_{t-1} \mid \mathbf{x}_t, \hat{\mathbf{x}}_0 = f_\theta(\mathbf{x}_t, t)), \quad (2)$$

where $\alpha_t \in [0, 1]$ is a decreasing sequence, which determines the variance schedule $\{\beta_t = 1 - \frac{\alpha_t}{\alpha_{t-1}}\}_{t=1}^T$, and $q(\mathbf{x}_{t-1} \mid \mathbf{x}_t, \mathbf{x}_0)$ is the conditional posterior of the forward diffusion chain that is also used to construct the reverse diffusion chain. A standout feature of this construction is that both the forward marginal and the reverse conditional are analytic, following the Gaussian distributions as

$$q(\mathbf{x}_t \mid \mathbf{x}_0) = \mathcal{N}(\mathbf{x}_t; \sqrt{\alpha_t}\mathbf{x}_0, (1 - \alpha_t)\mathbf{I}), \quad (3)$$

$$q(\mathbf{x}_{t-1} \mid \mathbf{x}_t, \mathbf{x}_0) = \mathcal{N}\left(\frac{\sqrt{\alpha_{t-1}}}{1-\alpha_t}(1 - \frac{\alpha_t}{\alpha_{t-1}})\mathbf{x}_0 + \frac{(1-\alpha_{t-1})\sqrt{\alpha_t}}{(1-\alpha_t)\sqrt{\alpha_{t-1}}}\mathbf{x}_t, \frac{1-\alpha_{t-1}}{1-\alpha_t}(1 - \frac{\alpha_t}{\alpha_{t-1}})\mathbf{I}\right). \quad (4)$$

Furthermore, maximizing the ELBO can be conducted by sampling $t \in \{1, \dots, T\}$ and minimizing

$$L_t = \text{KL}(q(\mathbf{x}_{t-1} \mid \mathbf{x}_t, \mathbf{x}_0) || q(\mathbf{x}_{t-1} \mid \mathbf{x}_t, \hat{\mathbf{x}}_0 = f_\theta(\mathbf{x}_t, t))) = \frac{\text{SNR}_{t-1} - \text{SNR}_t}{2}\|\mathbf{x}_0 - f_\theta(\mathbf{x}_t, t)\|_2^2, \quad (5)$$

where $\text{SNR}_t := \alpha_t/(1 - \alpha_t)$. From Equation 3, we can sample $\mathbf{x}_t$ via reparameterization as $\mathbf{x}_t = \sqrt{\alpha_t}\mathbf{x}_0 + \sqrt{1 - \alpha_t}\epsilon$, $\epsilon \sim \mathcal{N}(\mathbf{0}, \mathbf{I})$, and hence we can parameterize $\hat{\mathbf{x}}_0$ as a linear combination of $\mathbf{x}_t$ and a noise prediction $\epsilon_\theta(\mathbf{x}_t, t)$ as

$$\hat{\mathbf{x}}_0(\mathbf{x}_t, t; \theta) = f_\theta(\mathbf{x}_t, t) = \frac{\mathbf{x}_t}{\sqrt{\alpha_t}} - \frac{\sqrt{1-\alpha_t}}{\sqrt{\alpha_t}}\epsilon_\theta(\mathbf{x}_t, t). \quad (6)$$

Thus we can also express $L_t$ in Equation 5 in terms of noise prediction as $L_t = \frac{1}{2}(\frac{\text{SNR}_{t-1}}{\text{SNR}_t} - 1)\|\epsilon - \epsilon_\theta(\mathbf{x}_t, t)\|_2^2$. In summary, the network parameter $\theta$ can be trained by predicting either $\mathbf{x}_0$ (Karras et al., 2022) or the injected noise $\epsilon$ in forward diffusion (Ho et al., 2020; Song et al., 2021):

$$\theta^\star = \arg\min_\theta \mathbb{E}_{t,\mathbf{x}_t,\epsilon}[\lambda_t'\|\hat{\mathbf{x}}_0(\mathbf{x}_t, t; \theta) - \mathbf{x}_0\|_2^2], \text{ or } \theta^\star = \arg\min_\theta \mathbb{E}_{t,\mathbf{x}_t,\epsilon}[\lambda_t\|\epsilon_\theta(\mathbf{x}_t, t) - \epsilon\|_2^2], \quad (7)$$

where $\lambda_t$, $\lambda_t'$ are both time-dependent weight coefficients, which are often chosen to be different from the ones suggested by the ELBO to enhance image generation quality (Ho et al., 2020).

## 2.3 Stackable and Skippable LEGO Bricks: Training and Sampling

In our architectural design, we utilize the diffusion loss to train LEGO bricks at various levels of spatial granularity, with each level corresponding to specific spatial refinement steps. We employ

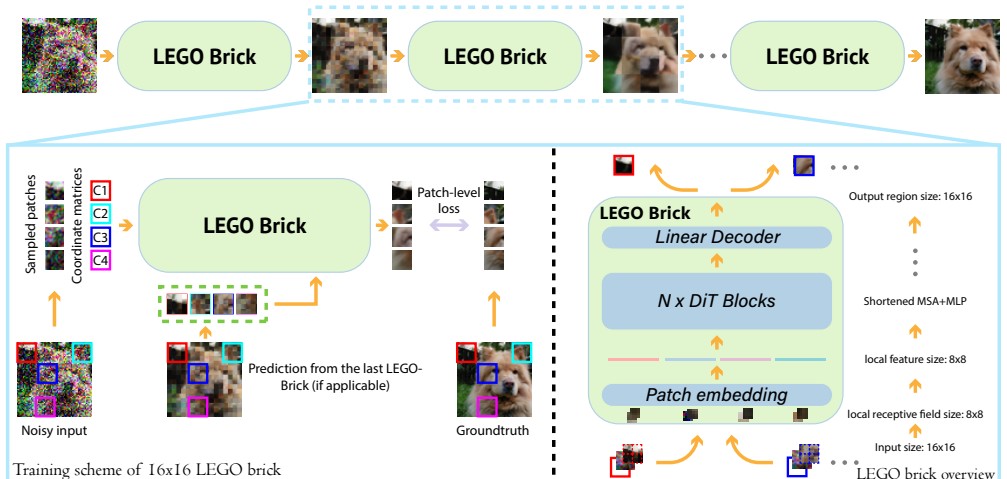

Figure 3: Top Panel: The generation paradigm with LEGO spatial refinement in a single time step. Bottom Panel: An overview of the training scheme (*Left*) and architecture (*Right*) of a LEGO brick, whose input size is $16 \times 16$, local receptive field size is $8 \times 8$, and attention span is 4.

Transformer blocks to capture global content at the patch level, aligning them with the specific LEGO brick in operation. Unlike U-Net, which convolves local filters across all pixel locations, our method enriches feature embeddings from non-overlapping local regions and does not include upsampling layers. In contrast to ViT, our approach operates many of its network layers at local patches of various sizes. This enables us to efficiently capture localized information and progressively aggregate it.

### 2.3.1 LEGO BRICK ENSEMBLE AND VERTICAL SPATIAL REFINEMENT

Diffusion models use their network backbone to iteratively refine image generation over the time dimension. When employing a backbone consisting of $K$ stacked LEGO bricks, we can progressively refine spatial features at each time step along the vertical dimension. When a LEGO brick processes its patches, it independently refines each patch within the same brick. Additionally, we maintain the full-resolution image consistently at both the input and output of all bricks. This not only provides flexibility in choosing both patch sizes and locations for various bricks but also creates a test-time reconfigurable structure that can selectively skip bricks to significantly reduce sampling costs.

Denote the original image with spatial dimensions $H \times W$ as $\mathbf{x}$. For the $k^{th}$ LEGO brick, which operates on patches of size $r_h(k) \times r_w(k)$, where $r_h(k) \leq H$ and $r_w(k) \leq W$, we extract a set of patches of that size from $\mathbf{x}$. To simplify, we assume the brick size $r_h(k) = r_w(k) = r_k$, both $\frac{H}{r_k}$ and $\frac{W}{r_k}$ are integers, and the image is divided into non-overlapping patches, represented as:

$$\mathbf{x}_{(i,j)}^{(k)} = \mathbf{x}[(i-1)r_k + 1 : ir_k, (j-1)r_k + 1 : jr_k]; \ i \in \{1, ..., \tfrac{H}{r_k}\}, \ j \in \{1, ..., \tfrac{W}{r_k}\}.$$

We also denote $\mathbf{m} \in [-1, 1]^{H \times W}$ as the normalized coordinates of the image pixels, and similarly, $\mathbf{m}_{(i,j)}^{(k)}$ as the coordinate matrix of the $(i,j)^{th}$ patch at the $k^{th}$ LEGO brick.

Mathematically, for the $k^{th}$ LEGO brick parameterized by $\theta_k$, and denoting $\mathbf{z}_{t,(i,j)}^{(k)}$ as the $(i,j)^{th}$ patch extracted at time $t$ from its output (with $\mathbf{z}_{t,(i,j)}^{(0)} = \emptyset$), the LEGO operation can be expressed as:

$$\mathbf{z}_{t,(i,j)}^{(k)} = f_{\theta_k}(\mathbf{x}_{t,(i,j)}^{(k)}, \mathbf{m}_{(i,j)}^{(k)}, \mathbf{z}_{t,(i,j)}^{(k-1)}, t), \tag{8}$$

which receives patches at the same spatial locations from both the previous time step and the lower brick as its input. For now, we treat this patch-level operation-based LEGO brick as a black box and will not delve into its details until we finish describing its recursive-based ensemble and training loss. To streamline the discussion, we will omit patch indices and coordinates when appropriate.

**Recursive ensemble of LEGO bricks:** The vanilla denoising diffusion step, as in Equation 6, is decomposed into $K$ consecutive LEGO bricks, stacked from the top to the bottom as follows:

$$\hat{\mathbf{x}}_0(\mathbf{x}_t, t; \theta) = \mathbf{z}_t^{(K)}, \ \text{where} \ \mathbf{z}_t^{(k)} = f_{\theta_k}(\mathbf{x}_t, \mathbf{z}_t^{(k-1)}, t) \ \text{for} \ k = K, \dots, 1, \tag{9}$$

with $\mathbf{z}_t^{(0)} := \emptyset$, $\theta := \{\theta_k\}_{1,K}$, and $\mathbf{z}_t^{(k)}$ denoting a grid of refined patches based on the corresponding patches from the output of the lower LEGO brick at time $t$. We note that since each LEGO brick starts with a full-resolution image and ends with a refined full-resolution image, we have the flexibility to choose the target brick size for each LEGO brick. In our illustration shown in Figure 3, we follow a progressive-growing-based construction, where the patch size of the stacked LEGO bricks monotonically increases when moving from lower to upper bricks. Alternatively, we can impose a monotonically decreasing constraint, corresponding to progressive refinement. In this work, we explore both progressive growth and refinement as two special examples, and further present a Unet-inspired structural variant that combines both. The combinatorial optimization of brick sizes for the LEGO bricks in the stack is a topic that warrants further investigation.

**Training loss:**  Denoting a noise-corrupted patch at time $t$ as $\mathbf{x}_t^{(k)}$, we have the diffusion chains as

$$\text{Forward}: q(\mathbf{x}_{0:T}^{(k)}) = q(\mathbf{x}_0^{(k)}) \prod_{t=1}^{T} \mathcal{N}\left(\mathbf{x}_t^{(k)}; \frac{\sqrt{\alpha_t}}{\sqrt{\alpha_{t-1}}}\mathbf{x}_{t-1}^{(k)}, 1 - \frac{\alpha_t}{\alpha_{t-1}}\right), \tag{10}$$

$$\text{Reverse}: p_\theta(\mathbf{x}_{0:T}) = p(\mathbf{x}_T) \prod_{t=1}^{T} q(\mathbf{x}_{t-1}^{(k)} \mid \mathbf{x}_t^{(k)}, \hat{\mathbf{x}}_0^{(k)} = f_\theta(\mathbf{x}_t^{(k)}, \mathbf{x}_t^{(k-1)}, t)), \tag{11}$$

Denote $\epsilon$ as a normal noise used to corrupt the clean image. The clean image patches, denoted as $\mathbf{x}_{0,(i,j)}^{(k)}$, are of size $r_k \times r_k$ and correspond to specific spatial locations $(i, j)$. Additionally, we have refined patches, $\hat{\mathbf{x}}_{0,(i,j)}^{(k)}$, produced by the $k^{th}$ LEGO brick at the same locations, and $\hat{\mathbf{x}}_{0,(i,j)}^{(k-1)}$ from the $(k-1)^{th}$ LEGO brick. When processing a noisy image $\mathbf{x}_t$ at time $t$, we perform upward propagation through the stacked LEGO bricks to progressively refine the full image. This begins with $\hat{\mathbf{x}}_0^{(0)} = \emptyset$ and proceeds with refined image patches $\hat{\mathbf{x}}_0^{(k)}$ for $k = 1, \ldots, K$. As each LEGO brick operates on sampled patches, when the number of patches is limited, there may be instances where $\hat{\mathbf{x}}_{0,(i,j)}^{(k-1)}$ contain missing values. In such cases, we replace these missing values with the corresponding pixels from $\mathbf{x}_0$ to ensure the needed $\hat{\mathbf{x}}_{0,(i,j)}^{(k-1)}$ is present. Denote $\lambda_t^{(k)}$ as time- and brick-dependent weight coefficients, whose settings are described in Appendix C. With the refined image patches $\hat{\mathbf{x}}_{0,(i,j)}^{(k)}$, we express the training loss over the $K$ LEGO bricks as

$$\mathbb{E}_k \mathbb{E}_{t,\mathbf{x}_0^{(k)}, \epsilon, (i,j)}[\lambda_t^{(k)} \|\mathbf{x}_{0,(i,j)}^{(k)} - \hat{\mathbf{x}}_{0,(i,j)}^{(k)}\|_2^2], \quad \hat{\mathbf{x}}_{0,(i,j)}^{(k)} := f_{\theta_k}(\mathbf{x}_{t,(i,j)}^{(k)}, \hat{\mathbf{x}}_{0,(i,j)}^{(k-1)}, t). \tag{12}$$

This training scheme enables each LEGO brick to refine local patches while being guided by the noise-corrupted input and the output of the previous brick in the stack.

### 2.3.2 Network Design within the LEGO Brick

In the design of a LEGO brick, we draw inspiration from the design of the Diffusion Transformer (DiT) proposed in Peebles & Xie (2022). The DiT architecture comprises several key components: Patch Embedding Layer: Initially, the input image is tokenized into a sequence of patch embeddings; DiT Blocks: These are Transformer blocks featuring multi-head attention, zero-initialized adaptive layer norm (adaLN-zero), and MLP layers; Linear layer with adaLN: Positioned on top of the DiT blocks, this linear layer, along with adaLN, converts the sequence of embeddings back into an image. The diffusion-specific conditions, such as time embedding and class conditions, are encoded within the embeddings through adaLN in every block.

As illustrated in Figure 3, we employ the DiT blocks, along with the patch-embedding layer and the linear decoder to construct a LEGO brick. The input channel dimension of the patch-embedding layer is expanded to accommodate the channel-wise concatenated input $[\mathbf{x}_t^{(k)}, \hat{\mathbf{x}}_0^{(k-1)}, \mathbf{m}^{(k)}]$. The size of the local receptive fields, denoted as $\ell_k \times \ell_k$, is subject to variation to achieve spatial refinement. In this paper, by default, we set it as follows: if the brick size $r_k \times r_k$ is smaller than the image resolution, it is set to $r_k/2 \times r_k/2$, and if the brick size is equal to the image resolution, it is set to $2 \times 2$. In simpler terms, for the $k^{th}$ LEGO brick, when $r_k \times r_k$ is smaller than $W \times H$ (or $W/8 \times H/8$ for diffusion in the latent space), we partition each of the input patches of size $r_k \times r_k$ into four non-overlapping local receptive fields of size $r_k/2 \times r_k/2$. These four local receptive fields are further projected by the same MLP to become a sequence of four token embeddings. Afterward, these four token embeddings are processed by the DiT blocks, which have an attention span as short as four, and decoded to produce output patches of size $r_k \times r_k$.

Table 1: FID results of unconditional image generation on CelebA $64 \times 64$ (Liu et al., 2015). All baselines use a Transformer-based backbone.

| Method | #Params | FLOPs | FID |
|---|---|---|---|
| U-ViT-S/4 | 44M | 1.2G | 2.87 |
| DiT-S/2 | 33M | 0.4G | 2.52 |
| LEGO-S-PG (ours) | 35M | 0.2G | 2.17 |
| LEGO-S-PR (ours) | 35M | 0.2G | **2.09** |

Table 2: FID results of conditional image generation on ImageNet $64 \times 64$ (Deng et al., 2009). All baselines use a Transformer-based backbone.

| Method | #Params | FLOPs | FID |
|---|---|---|---|
| U-ViT-L/4 | 287M | 91.2G | 4.26 |
| DiT-L/2 | 458M | 161.4G | 2.91 |
| LEGO-L-PG (ours) | 464M | 68.8G | **2.16** |
| LEGO-L-PR (ours) | 464M | 68.8G | 2.29 |

We categorize the LEGO bricks as 'patch-bricks' if their brick size is smaller than the image resolution, and as 'image-bricks' otherwise. The patch-bricks have attention spans as short as four, saving memory and computation in both training and generation. While the image-bricks choose longer attention spans and hence do not yield memory or computation savings during training, they can be compressed with fewer DiT blocks compared with the configuration in Peebles & Xie (2022). During generation, LEGO bricks can also be selectively skipped at appropriate time steps without causing clear performance degradation, resulting in substantial savings in generation costs. For simplicity, we maintain a fixed embedding size and vary only the number of DiT blocks in each brick based on $r_k$. Each LEGO brick may require a different capacity depending on $r_k$. For a comprehensive configuration of the LEGO bricks, please refer to Table 5 in Appendix C.

We explore three spatial refinement settings and stack LEGO bricks accordingly: 1) Progressive Growth (PG): In this setup, we arrange LEGO bricks in a manner where each subsequent brick has a patch size that's four times as large as the previous one, $i.e.$, $r_k = 4r_{k-1}$. Consequently, the patch of the current break would aggregate the features of four patches output by the previous brick, facilitating global-content orchestration. 2) Progressive Refinement (PR): By contrast, for the PR configuration, we stack LEGO bricks in reverse order compared to PG, with $r_k = r_{k-1}/4$. Here, each LEGO brick is responsible for producing a refined generation based on the content provided by the brick below it. 3) LEGO-U: a hierarchical variant that combines the features of LEGO-PR and LEGO-PG. This model processes image patches starting with larger resolutions, transitioning to smaller ones, and then reverting back to larger resolutions, similar to Unets that operate with multiple downsampling and upsampling stages to have multi-scale representations.

## 3 EXPERIMENTS

We present a series of experiments designed to assess the efficacy and versatility of using stacked LEGO bricks as the backbone for diffusion models. We defer the details on datasets, training settings, and evaluation metrics into Appendix C. We begin by showcasing results on both small-scale and large-scale datasets to evaluate the image generation performance using the three configurations outlined in Table 5 in Appendix C.

### 3.1 MAIN RESULTS

**Training in pixel space:** We begin by comparing LEGO in pixel-space training on CelebA and ImageNet with two Transformer-based state-of-the-art diffusion models (Bao et al., 2022; Peebles & Xie, 2022), which are our most relevant baselines. All images are resized to a resolution of $64 \times 64$. We conduct experiments with two spatial refinement methods using LEGO, denoted as -PG for the progressive growth variant and -PR for the progressive refinement variant, as explained in Section 2.3.2. In both Tables 1 and 2, when compared to DiT, LEGO slightly increases the number of parameters due to the expansion of the input dimension in a channel-wise manner. However, in terms of FLOPs, we observe that LEGO incurs a significantly lower computational cost compared to both U-ViT and DiT. Despite maintaining such a low computational cost, LEGO still wins the competition in terms of FID performance.

**Training in latent space:** We conduct comparisons with state-of-the-art class-conditional generative models trained on ImageNet at resolutions of $256 \times 256$ and $512 \times 512$, as shown in Table 3. Our results are obtained after training with 512M images (calculated as the number of iterations $\times$ batch size). When considering LEGO without classifier-free guidance (CFG), we observe that it achieves a better FID than the baseline diffusion models, striking a good balance between precision and

Table 3: Comparison of LEGO-diffusion with state-of-the-art generative models on class-conditional generation using ImageNet at resolutions 256×256 and $512 \times 512$. Each metric is presented in two columns, one without classifier-free guidance and one with, marked with ✗ and ✓, respectively.

| Evaluation Metric | FID↓ | | sFID↓ | | IS↑ | | Prec.↑ | | Rec.↑ | | Iterated |
|---|---|---|---|---|---|---|---|---|---|---|---|
| Classifier-free guidance | ✗ | ✓ | ✗ | ✓ | ✗ | ✓ | ✗ | ✓ | ✗ | ✓ | Images (M) |
| *U-Net-based architecture (256 × 256)* | | | | | | | | | | | |
| ADM (Dhariwal & Nichol, 2021) | 10.94 | 4.59 | 6.02 | 5.25 | 100.98 | 186.70 | 0.69 | 0.82 | 0.63 | 0.52 | 506 |
| ADM-Upsampled | 7.49 | 3.94 | 5.13 | 6.14 | 127.49 | 215.84 | 0.72 | 0.83 | 0.63 | 0.53 | |
| LDM-8 (Rombach et al., 2022) | 15.51 | 7.76 | - | - | 79.03 | 103.49 | 0.65 | 0.71 | 0.63 | **0.62** | 307 |
| LDM-4 | 10.56 | 3.60 | - | - | 103.49 | 247.67 | 0.71 | **0.87** | 0.62 | 0.48 | 213 |
| *Transformer-based architecture (256 × 256)* | | | | | | | | | | | |
| U-ViT-H/2 (Bao et al., 2022) | 6.58 | 2.29 | - | 5.68 | - | 263.88 | - | 0.82 | - | 0.57 | 307 |
| DiT-XL/2 (Peebles & Xie, 2022) | 9.62 | 2.27 | 6.85 | 4.60 | 121.50 | 278.24 | 0.67 | 0.83 | **0.67** | 0.57 | 1792 |
| MDT-XL/2 (Gao et al., 2023) | 6.23 | **1.79** | 5.23 | 4.57 | 143.02 | 283.01 | 0.71 | 0.81 | 0.65 | 0.6 | 1664 |
| MaskDiT (Zheng et al., 2023) | 5.69 | 2.28 | 10.34 | 5.67 | 177.99 | 276.56 | 0.74 | 0.80 | 0.60 | 0.61 | 521 |
| LEGO-XL-PG (ours, 256 × 256) | 7.15 | 2.05 | 7.71 | 4.77 | 192.75 | 289.12 | **0.77** | 0.84 | 0.63 | 0.55 | 512 |
| LEGO-XL-PR (ours, 256 × 256) | 5.38 | 2.35 | 9.06 | 5.21 | 189.12 | 284.73 | **0.77** | 0.83 | 0.63 | 0.60 | 512 |
| LEGO-XL-U (ours, 256 × 256) | **5.28** | 2.59 | **4.91** | **4.21** | **337.85** | **338.08** | 0.67 | 0.83 | 0.65 | 0.56 | 350 |
| *U-Net-based architecture (512 × 512)* | | | | | | | | | | | |
| ADM (Dhariwal & Nichol, 2021) | 23.24 | 7.72 | 10.19 | 6.57 | 58.06 | 172.71 | 0.73 | **0.87** | 0.60 | 0.42 | 496 |
| ADM-Upsampled | 9.96 | 3.85 | **5.62** | 5.86 | 121.78 | 221.72 | 0.75 | 0.84 | **0.64** | 0.53 | |
| *Transformer-based architecture (512 × 512)* | | | | | | | | | | | |
| DiT-XL/2 (Peebles & Xie, 2022) | 12.03 | **3.04** | 7.12 | 5.02 | 105.25 | 240.82 | 0.75 | 0.84 | **0.64** | 0.54 | 768 |
| LEGO-XL-PG (ours, 512 × 512) | 10.06 | 3.74 | 5.96 | **4.62** | **192.27** | **285.66** | **0.77** | 0.85 | **0.64** | **0.64** | 512 |
| LEGO-XL-PR (ours, 512 × 512) | **9.01** | 3.99 | 6.39 | 4.87 | 168.92 | 265.75 | 0.76 | 0.86 | 0.63 | 0.49 | 512 |

recall. One potential explanation for this improvement is that the LEGO bricks with small patch sizes enhance patch diversity, preserving both generation fidelity and diversity.

With the inclusion of CFG, we notably achieve an FID of 2.05 and the IS of 289.12 on ImageNet-$256 \times 256$. It is notable that the LEGO-U variant, which incorporates the features of the -PR and -PG variants, has achieved a clearly higher IS (338.08 *w.* guidance and 337.85 *w.o.* guidance). This result significantly surpasses the baselines and demonstrates the power of multi-scale information in generative modeling. In the case of using $512 \times 512$ resolution data, LEGO also achieves the best IS and produces a competitive FID with fewer iterated images. Additionally, as depicted in Figure 2, LEGO demonstrates superior convergence speed and requires significantly less training time. For qualitative validation, we provide visualizations of randomly generated samples in Figure 1.

## 3.2 IMPROVING THE EFFICIENCY OF DIFFUSION MODELING WITH LEGO

The design of LEGO inherently facilitates the sampling process by generating images with selected LEGO bricks. Intuitively, at low-noise timesteps (*i.e.*, when $t$ is small), the global structure of images is already well-defined, and hence the model is desired to prioritize local details. Conversely, when images are noisier, the global-content orchestration becomes crucial to uncover global structure under high uncertainties. Therefore, for improved efficiency, we skip LEGO bricks that emphasize local details during high-noise timesteps and those that construct global structures during low-noise timesteps. To validate this design, we conduct experiments to study the generation performance when LEGO bricks are skipped in different degrees.

For LEGO-PG, we designate a timestep $t_{break}$ as a breakpoint. When $t > t_{break}$, no bricks are skipped, but when $t \leq t_{break}$, the top-level brick, which is the image-brick with a brick size of $64 \times 64$, is skipped for the remainder of the sampling process. In contrast, for PR, we set a timestep $t_{break}$ as a breakpoint. When $t \leq T - t_{break}$, no bricks are skipped, but when $t > T - t_{break}$, the top-level brick, which is the patch-break with a size of $4 \times 4$, is skipped for the remainder of the sampling process.

We evaluate the impact of the choice of $t_{break}$ on both FID scores and sampling time for LEGO-PG and LEGO-PR. The results in Figure 4 illustrate the FID scores and the time required to generate 50,000 images using 8 NVIDIA A100 GPUs. In LEGO-PG, when we skip the top-level LEGO brick, we observe a notable enhancement in sampling efficiency, primarily because this brick, responsible for processing the entire image, is computationally demanding. Nevertheless, skipping this brick

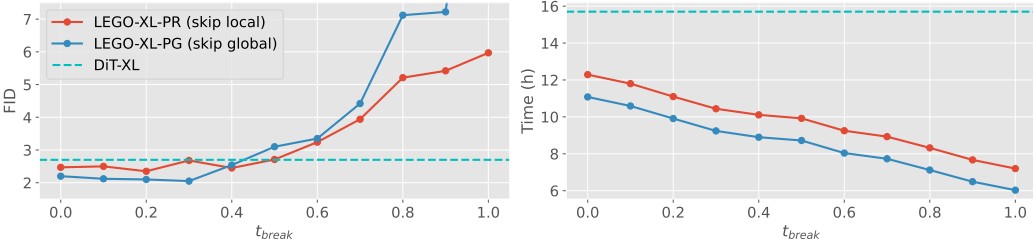

Figure 4: Visualization of how FID and inference time for 50k images change as the proportion of reverse diffusion time steps, at which the top-level brick is skipped, increases from 0 to 1. LEGO-PG prioritizes skipping the top-level brick at earlier time steps, while LEGO-PR prioritizes it at later time steps. The experiments are conducted using the LEGO-XL model with eight NVIDIA A100 GPUs on the ImageNet ($256 \times 256$) dataset.

during timesteps with high levels of noise also results in a trade-off with performance, as the model loses access to global content. On the other hand, for LEGO-PR, skipping the top-level patch-brick slightly impacts performance when $t_{\text{break}}$ is large, while the improvement in sampling efficiency is not as substantial as with LEGO-PG. Interestingly, when $t_{\text{break}}$ is chosen to be close to the halfway point of sampling, performance is preserved for both models, and significant time savings can be achieved during sampling.

### 3.3 LEVERAGING A PRETRAINED DIFFUSION MODEL AS A LEGO BRICK

Apart from the training efficiency, another advantage of the design of LEGO is any pre-trained diffusion models can be incorporated as a LEGO brick as long as they can predict $\hat{\mathbf{x}}_0^{(k)}$. A natural choice is to deploy a pre-trained diffusion model as the first LEGO brick in LEGO-PR. Then, a series of LEGO bricks that refine local details can be built on top of it. To validate this concept, we conducted experiments on CIFAR-10 using LEGO-PR. We deploy an unconditional Variance Preservation (VP) diffusion model (Song et al., 2021) (implemented under the EDM codebase provided by Karras et al. (2022)) to replace the LEGO brick that deals with full images, and build a LEGO-S ($4 \times 4$) brick on the top for refinement. We train the incorporated model in two different ways: 1) VP (frozen) + LEGO-PR: The VP model is fixed and only the other LEGO bricks are trained, and 2) VP (unfrozen) + LEGO-PR: The VP model is fine-tuned along with the training of the other LEGO bricks. Table 4 shows that the combined model consistently improves the base model to obtain a better generation result, demonstrating the versatility of LEGO bricks.

Table 4: Outcomes from CIFAR10 on integrating a pre-trained model as the first brick in LEGO-PR.

| Model | VP | VP (frozen) + LEGO-PR | VP (unfrozen) + LEGO-PR |
|---|---|---|---|
| FID | 2.5 | 1.92 | 1.88 |

## 4 CONCLUSION

We introduce the LEGO brick, a novel network unit that seamlessly integrates Local-feature Enrichment and Global-content Orchestration. These bricks can be stacked to form the backbone of diffusion models. LEGO bricks are adaptable, operating on local patches of varying sizes while maintaining the full-resolution image across all bricks. During training, they can be flexibly stacked to construct diffusion backbones of diverse capacity, and during testing, they are skippable, enabling efficient image generation with a smaller network than used in training. This design choice not only endows LEGO with exceptional efficiency during both training and sampling but also empowers it to generate images at resolutions significantly higher than training. Our extensive experiments, conducted on challenging image benchmarks such as CelebA and ImageNet, provide strong evidence of LEGO's ability to strike a compelling balance between computational efficiency and generation quality. LEGO accelerates training, expedites convergence, and consistently maintains, if not enhances, the quality of generated images. This work signifies a noteworthy advancement in addressing the challenges associated with diffusion models, making them more versatile and efficient for generating high-resolution photo-realistic images.

ACKNOWLEDGMENTS

H. Zheng, Z. Wang, and M. Zhou acknowledge the support of NSF-IIS 2212418, NIH-R37 CA271186, and the NSF AI Institute for Foundations of Machine Learning (IFML).

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

# A    DISCUSSIONS ON RELATED WORK

## MORE EFFICIENT DIFFUSION MODELS

Diffusion models have garnered increasing attention from researchers seeking to enhance them in various ways. These efforts include combining variational loss with weighted denoising loss (Dhariwal & Nichol, 2021; Kingma et al., 2021; Karras et al., 2022), accelerating sampling speed using ODE solvers (Kong & Ping, 2021; San-Roman et al., 2021; Song et al., 2020; Zhang & Chen, 2022; Lu et al., 2022), and employing auxiliary models to improve training (Rombach et al., 2022; Li et al., 2022b; Zheng et al., 2022c; Pandey et al., 2022). Additionally, the iterative refinement property of diffusion models has inspired research in various domains, such as stable GAN training (Xiao et al., 2022; Wang et al., 2022), large-scale text-to-image models (Kim & Ye, 2021; Nichol et al., 2021; Gu et al., 2022; Saharia et al., 2022), and robust super-resolution and editing models (Meng et al., 2021; Choi et al., 2021; Li et al., 2022a), among others.

With rapid advancements, diffusion models have often outperformed state-of-the-art generative adversarial networks (GANs, (Goodfellow et al., 2014)). GANs typically require adversarial training and are susceptible to issues such as unstable training and mode collapse. In contrast, diffusion models offer a versatile framework that is not bound by the traditional constraints associated with autoregressive models (Bengio & Bengio, 1999; Uria et al., 2013; 2016; Van Den Oord et al., 2016), variational autoencoders (VAEs, (Kingma & Welling, 2014; Rezende et al., 2014)), or flow-based models (Dinh et al., 2015; 2017; Kingma & Dhariwal, 2018; Song et al., 2023).

While diffusion models have consistently demonstrated impressive performance, they face challenges related to substantial training and inference overheads, especially when applied to large high-resolution image datasets. Much of the research on diffusion models has focused on improving their sampling efficiency. Some studies (Song et al., 2020; Lu et al., 2022; Karras et al., 2022) have refined sampling strategies and utilized sophisticated numerical solvers to enhance efficiency. In parallel, alternative approaches (Salimans & Ho, 2022; Meng et al., 2022; Zheng et al., 2022a; Song et al., 2023) have employed surrogate networks and distillation techniques. However, these solutions primarily target sampling efficiency and have not made substantial progress in reducing the training costs associated with diffusion models.

To address the challenge of training efficiency, latent-space diffusion models were introduced to transform high-resolution images into a manageable, low-dimensional latent space (Vahdat et al., 2021; Rombach et al., 2022). Ho et al. (2022) introduced a cascaded diffusion model approach, consisting of a foundational diffusion model for low-resolution images, followed by super-resolution ones. Similarly, Wang et al. (2023) proposed a segmented training paradigm, conducting score matching at a patch level. While this U-Net-based approach exhibits certain resemblances to ours, wherein a set of smaller patches serves as input for the Transformer blocks, notable distinctions exist.

Recently, Bao et al. (2022; 2023) and Peebles & Xie (2022) explored the use of Transformer architecture for diffusion models instead of the conventional U-Net architecture. Additionally, masked Transformers were utilized to improve the training convergence of diffusion models (Gao et al., 2023; Zheng et al., 2022a). However, that approach, which involves processing both masked and unmasked tokens, substantially increases the training overhead per iteration. In contrast, our method prioritizes efficiency by exclusively utilizing the unmasked tokens during training.

## BACKBONE OF DIFFUSION MODELS

A crucial element within diffusion models revolves around the network architecture employed in their iterative refinement-based generative process. The initial breakthrough, which propelled score- or diffusion-based generative models into the spotlight, can be attributed to the integration of the convolutional operation-based U-Net architecture (Song & Ermon, 2019; Ho et al., 2020). Originally conceived by Ronneberger et al. (2015) for biomedical image segmentation, the U-Net design incorporates both a progressive downsampling path and a progressive upsampling path, enabling it to generate images of the same dimensions as its input. In the context of diffusion models, the U-Net often undergoes further modifications, such as the incorporation of attention blocks, residual blocks, and adaptive normalization layers (Dhariwal & Nichol, 2021).

While the downsampling and upsampling convolutional layers in U-Net allow it to effectively capture local structures, achieving a good understanding of global structures often requires a deep architecture and self-attention modules embodied in the architecture, resulting in high computational complexity during both training and generation. Instead of depending on a large number of convolutional layers, a classical approach in signal and image processing involves the extraction of local feature descriptors and their application in downstream tasks (Lowe, 2004; Mikolajczyk & Schmid, 2005; Tuytelaars & Mikolajczyk, 2008). For instance, techniques like dictionary learning and sparse coding operate on the principle that each local patch can be expressed as a sparse linear combination of dictionary atoms learned from image patches. This learned patch dictionary can then be used to enrich all overlapping patches for various image-processing tasks, such as denoising and inpainting (Aharon et al., 2006; Mairal et al., 2007; Zhou et al., 2009).

The emergence of ViT in image classification (Dosovitskiy et al., 2021), which integrates extracting feature embeddings from non-overlapping local patches with the global attention mechanism of the Transformer, marks a resurgence of this classical local-feature-based concept. ViT has not only competed effectively with state-of-the-art convolution-based networks in both image classification (Touvron et al., 2022) and generation (Esser et al., 2021), but has also been adapted to replace the U-Net in diffusion models (Yang et al., 2022; Bao et al., 2022; Peebles & Xie, 2022; Gao et al., 2023), achieving state-of-the-art image generation performance. This underscores the notion that the convolutional architecture can be supplanted by the enrichment of local features and their orchestration through global attention. ViT-based models start by enriching patch-level representations and then utilize a sequence of Transformer blocks to capture dependencies among local patches. They often demand smaller patch sizes to capture fine-grained local details and a large number of attention layers to capture global dependencies, which results in a demand of enormous data to fit. This necessitates lengthy sequence computations and deep networks to achieve optimal results, thereby incurring significant computational costs. To enhance efficiency and expedite convergence, recent works have explored hybridizing the entangled global-local structure and multi-scale properties from convolutional networks into hierarchical transformers (Liu et al., 2021; Zheng et al., 2022b; Liu et al., 2022; Yu et al., 2024).

To bridge the gap between the convolutional U-Nets and the Transformer architecture, LEGO tackles the diffusion modeling from a view of local-feature enrichment and global-content orchestration. Inspired by convolutional networks, LEGO enriches local features at both local and global scales. However, different from a series of downsampling and upsampling stages used in the U-Nets, LEGO leverages varied patch decomposition methods. This distinction results in diverse feature representations. To achieve a global-content orchestration, LEGO deploys self-attention, whose length varies depending on the 'LEGO brick'. In some cases, attention is confined to specific image patches defined by the brick size, while in some other bricks, it extends over the entire image. This selective mechanism allows for more focused and efficient feature aggregation.

## B    LIMITATIONS AND FUTURE WORK

The current work exhibits several limitations. Firstly, we have not explored the application of LEGO bricks for text-guided image generation. Secondly, in generating class-conditional images at resolutions much higher than the training data, categories with low diversity may produce results that lack photorealism. Thirdly, our primary focus has been on stacking LEGO bricks either in a progressive-growth or progressive-refinement manner, with other stacking strategies remaining relatively unexplored. Fourthly, our selection of which LEGO bricks to skip at different time steps during generation relies on heuristic criteria, and there is room for developing more principled skipping strategies.

While the use of LEGO bricks has led to considerable computational savings in both training and generation, it still requires state-of-the-art GPU resources that are very expensive to run and hence typically beyond the reach of budget-sensitive projects. For example, training LEGO diffusion with 512M images on ImageNet $512 \times 512$ took 32 NVIDIA A100 GPUs with about 14 days.

It's important to note that like other diffusion-based generative models, LEGO diffusion can potentially be used for harmful purposes when trained on inappropriate image datasets. Addressing this concern is a broader issue in diffusion models, and LEGO bricks do not appear to inherently include mechanisms to mitigate such risks.

## C EXPERIMENTAL SETTINGS

**Datasets:** We conduct experiments using the CelebA dataset (Liu et al., 2015), which comprises 162,770 images of human faces, and the ImageNet dataset (Deng et al., 2009), consisting of 1.3 million natural images categorized into 1,000 classes. For data pre-processing, we follow the convention to first apply a center cropping and then resize it to $64 \times 64$ (Song et al., 2021) on CelebA; on ImageNet, we pre-process images at three different resolutions: $64 \times 64$, $256 \times 256$, and $512 \times 512$.

For images of resolution $64 \times 64$, we directly train the model in the pixel space. For higher-resolution images, we follow Rombach et al. (2022) to first encode the images into a latent space with a down-sampling factor of 8 and then train a diffusion model on that latent space. Specifically, we employ the EMA checkpoint of autoencoder from Stable Diffusion[1] to pre-process the images of size $256 \times 256 \times 3$ into $32 \times 32 \times 4$, and those of size $512 \times 512 \times 3$ into $64 \times 64 \times 4$.

**Training settings:** For our LEGO model backbones, we employ EDM (Karras et al., 2022) pre-conditioning on pixel-space and iDDPM (Nichol & Dhariwal, 2021; Dhariwal & Nichol, 2021) on latent-space data, strictly adhering to the diffusion designs prescribed by these respective preconditioning strategies.

During training, we employ the AdamW optimizer (Loshchilov & Hutter, 2018) for all experiments. For DDPM preconditioning, we use a fixed learning rate of $1 \times 10^{-4}$; for EDM preconditioning, we adopt the proposed learning rate warmup strategy, where the learning rate increases linearly to $1 \times 10^{-4}$ until iterated with 10k images. The batch size per GPU is set at 64 for all experiments. When trained on CelebA, we finish the training when the model is iterated with 2M images, and when trained on ImageNet, including $64 \times 64$, $256 \times 256$, and $512 \times 512$ resolution, we finish the training when the model is iterated with 512M images.

**Evaluation:** For performance evaluation, we primarily use FLOPs to measure the generation cost and generate 50,000 random images to compute the FID score as a performance metric. Additionally, we include the metrics (Ding et al., 2022; Kynkäänniemi et al., 2019) employed by Peebles & Xie (2022) for a comprehensive comparison. In the context of image generation, we employ the Heun sampler when using EDM preconditioning (Karras et al., 2022) and the DDPM sampler when utilizing iDDPM preconditioning (Nichol & Dhariwal, 2021).

Specifically, we employed the same suite, namely the ADM Tensorflow evaluation suite implemented in Dhariwal & Nichol (2021)[2]. We also utilized their pre-extracted features for the reference batch to ensure that our evaluations are rigorous and reliable. Additionally, we tested the evaluation with scripts and reference batch provided in EDM (Karras et al., 2022) [3]. We observed that the results from both evaluation suites were consistent, showing no significant differences (up to two decimal places).

**Diffusion settings** We deploy our LEGO model backbones with two commonly-used parameterization, namely DDPM (Nichol & Dhariwal, 2021; Dhariwal & Nichol, 2021) and EDM (Karras et al., 2022). Recall the training loss of diffusing models in Equation 7:

$$\theta^\star = \arg\min_\theta \mathbb{E}_{t,\mathbf{x}_t,\epsilon}[\lambda_t \|\epsilon_\theta(\mathbf{x}_t, t) - \epsilon\|_2^2], \text{ or } \theta^\star = \arg\min_\theta \mathbb{E}_{t,\mathbf{x}_t,\epsilon}[\lambda_t' \|\hat{\mathbf{x}}_0(\mathbf{x}_t, t; \theta) - \mathbf{x}_0\|_2^2].$$

The DDPM preconditioning follows the former equation, training the model to predict the noise injected to $\mathbf{x}_t$. Following Peebles & Xie (2022), we use the vanilla DDPM (Ho et al., 2020) loss, where $\lambda_t = 1$, and use the linear diffusion schedule. The EDM preconditioning can be regarded as an SDE model optimized using the latter equation, where the output is parameterized with:

$$\hat{\mathbf{x}}_0(\mathbf{x}_t, t; \theta) = f_\theta(\mathbf{x}_t, t; \theta) = c_{\text{skip}}(\sigma)\mathbf{x}_t + c_{\text{out}}(\sigma)\epsilon_\theta(c_{\text{in}}(\sigma)\mathbf{x}_t; c_{\text{noise}}(\sigma)),$$

where $\sigma$ depends on the noise-level of the current timestep. $c_{\text{skip}}(\sigma)$, $c_{\text{in}}$, $c_{\text{out}}$, and $c_{\text{noise}}$ are 4 hyper-parameters depending on $\sigma$. We strictly follow EDM to set these parameters and use the EDM schedule. More details regarding the diffusion settings can be found in Ho et al. (2020) and Karras et al. (2022).

---

[1] https://huggingface.co/stabilityai/sd-vae-ft-mse-original
[2] https://github.com/openai/guided-diffusion/tree/main/evaluations
[3] https://github.com/nvlabs/edm

Table 5: LEGO-Diffusion model architectures with different configurations. Following the convention, we introduce three different configurations—Small, Large, and XLarge—for different model capacities. The generation resolution is assumed to be $64 \times 64$. In each LEGO brick, $l$ and $d$ represent the size of local receptive fields and their embedding dimension, respectively. The number of tokens (attention span) in a LEGO brick is determined as the brick size (input & output patch size) $r \times r$ divided by its local-receptive field size $\ell \times \ell$. LEGO-PG stacks the three LEGO bricks with brick sizes of $4 \times 4$, $16 \times 16$, and $64 \times 64$ from the bottom to the top to form the network backbone for diffusion modeling. By contrast, LEGO-PR stacks them in the reverse order.

| Brick Size | Layer Name | LEGO-S | LEGO-L | LEGO-XL |
|---|---|---|---|---|
| | Token Embedding | $\ell = 2; d = 384$ | $\ell = 2; d = 1024$ | $\ell = 4; d = 1152$ |
| $4 \times 4$ | LEGO patch-brick | $\begin{bmatrix} \text{\# attention heads} = 6 \\ \text{mlp ratio} = 4 \end{bmatrix} \times 2$ | $\begin{bmatrix} \text{\# attention heads} = 16 \\ \text{mlp ratio} = 4 \end{bmatrix} \times 4$ | $\begin{bmatrix} \text{\# attention heads} = 16 \\ \text{mlp ratio} = 4 \end{bmatrix} \times 4$ |
| | Token Embedding | $\ell = 8; d = 384$ | $\ell = 8; d = 1024$ | $\ell = 8; d = 1152$ |
| $16 \times 16$ | LEGO patch-brick | $\begin{bmatrix} \text{\# attention heads} = 6 \\ \text{mlp ratio} = 4 \end{bmatrix} \times 4$ | $\begin{bmatrix} \text{\# attention heads} = 16 \\ \text{mlp ratio} = 4 \end{bmatrix} \times 8$ | $\begin{bmatrix} \text{\# attention heads} = 16 \\ \text{mlp ratio} = 4 \end{bmatrix} \times 12$ |
| | Token Embedding | $\ell = 2; d = 384$ | $\ell = 2; d = 1024$ | $\ell = 2; d = 1152$ |
| $64 \times 64$ | LEGO image-brick | $\begin{bmatrix} \text{\# attention heads} = 6 \\ \text{mlp ratio} = 4 \end{bmatrix} \times 6$ | $\begin{bmatrix} \text{\# attention heads} = 16 \\ \text{mlp ratio} = 4 \end{bmatrix} \times 12$ | $\begin{bmatrix} \text{\# attention heads} = 16 \\ \text{mlp ratio} = 4 \end{bmatrix} \times 14$ |
| Parameter size (M) | | 35 | 464 | 681 |

Table 6: Analogous to Table 5 for LEGO to model $32 \times 32$ resolution.

| Brick Size | Layer Name | LEGO-S | LEGO-L | LEGO-XL |
|---|---|---|---|---|
| | Token Embedding | $\ell = 2; d = 384$ | $\ell = 2; d = 1024$ | $\ell = 4; d = 1152$ |
| $4 \times 4$ | LEGO patch-brick | $\begin{bmatrix} \text{\# attention heads} = 6 \\ \text{mlp ratio} = 4 \end{bmatrix} \times 2$ | $\begin{bmatrix} \text{\# attention heads} = 16 \\ \text{mlp ratio} = 4 \end{bmatrix} \times 4$ | $\begin{bmatrix} \text{\# attention heads} = 16 \\ \text{mlp ratio} = 4 \end{bmatrix} \times 4$ |
| | Token Embedding | $\ell = 4; d = 384$ | $\ell = 4; d = 1024$ | $\ell = 4; d = 1152$ |
| $8 \times 8$ | LEGO patch-brick | $\begin{bmatrix} \text{\# attention heads} = 6 \\ \text{mlp ratio} = 4 \end{bmatrix} \times 4$ | $\begin{bmatrix} \text{\# attention heads} = 16 \\ \text{mlp ratio} = 4 \end{bmatrix} \times 8$ | $\begin{bmatrix} \text{\# attention heads} = 16 \\ \text{mlp ratio} = 4 \end{bmatrix} \times 12$ |
| | Token Embedding | $\ell_3 = 2; d = 384$ | $\ell_3 = 2; d = 1024$ | $\ell_3 = 2; d = 1152$ |
| $32 \times 32$ | LEGO image-brick | $\begin{bmatrix} \text{\# attention heads} = 6 \\ \text{mlp ratio} = 4 \end{bmatrix} \times 6$ | $\begin{bmatrix} \text{\# attention heads} = 16 \\ \text{mlp ratio} = 4 \end{bmatrix} \times 12$ | $\begin{bmatrix} \text{\# attention heads} = 16 \\ \text{mlp ratio} = 4 \end{bmatrix} \times 14$ |
| Parameter size (M) | | 35 | 464 | 681 |

During generation, for DDPM preconditioning, we use the diffusion sampler for 250 steps with a uniform stride, proposed in Nichol & Dhariwal (2021). For EDM preconditioning, we use the $2^{nd}$ order Heun sampler in Karras et al. (2022). On CelebA, we sample 75 steps with the deterministic sampler. On Imagenet, we sample for 256 steps with the stochastic sampler. The stochasticity-related hyper-parameters are set as: $S_{\text{churn}} = 10$, $S_{\text{min}} = 0.05$, $S_{\text{max}} = 20$, and $S_{\text{noise}} = 1.003$.

**Model design** We provide an overview of the architecture configurations for the LEGO models with different capacities in Tables 5 and 6. Since Table 6 only differs from Table 5 in the brick sizes and local receptive field sizes, we will focus on explaining Table 5 in detail. Table 5 consists of three LEGO bricks, with two of them being patch-level bricks and one being an image-level brick. When using the PG-based spatial refinement approach, we stack the bricks from the bottom to the top following the order: $4 \times 4 \rightarrow 16 \times 16 \rightarrow 64 \times 64$. In other words, the $4 \times 4$ patch-level brick is placed at the bottom, followed by the $16 \times 16$ patch-level brick, and then the $64 \times 64$ image-level brick. Conversely, for the PR-based spatial refinement approach, we stack the bricks from the bottom to the top following the order: $64 \times 64 \rightarrow 16 \times 16 \rightarrow 4 \times 4$. Adhering to conventional modeling techniques, we've outlined three distinct configurations — Small (LEGO-S), Large (LEGO-L), and XLarge (LEGO-XL) — used in our main experiments to cater to varying model capacities. Across bricks, the main differences are in token embedding and the number of attention heads, which revolves around the feature size of a local receptive field. Each brick specifies its patch size $r_k \times r_k$, local receptive field size $\ell_k \times \ell_k$, and embedding dimension (number of channels) $d_k$. The number of tokens (attention span) is determined as $(r_k/\ell_k)^2$. In each brick, the configurations for LEGO-S,

LEGO-L, and LEGO-XL also differ in terms of the number of attention heads and the number of DiT blocks used in the current brick to ensure scalability and adaptability across the architecture. To maintain generation coherence, we set the local-receptive-field size of the image-level brick as $2 \times 2$, which effectively turns it into a DiT with a long attention span, albeit with significantly fewer layers compared to standard DiT models. Moreover, the image-level brick can be safely skipped at appropriate time steps during generation. We find the proposed configuration strikes a favorable balance between computational efficiency and performance, which is comprehensively studied in our ablation studies on the effects of LEGO bricks in Appendix D.2. During training, the LEGO patch-bricks can operate on sampled patches to further save memory and computation. In our experiments on the CelebA and ImageNet datasets, we utilize 50% and 75% of all non-overlapping patches for training patch-bricks, respectively. As noted in Section 2.2, there are instances where $\hat{\mathbf{x}}_{0,(i,j)}^{(k-1)}$ contains missing values as some patches are not sampled in the preceding brick. In such cases, we replace these missing values with the corresponding pixels from $\mathbf{x}_0$ to ensure that the required $\hat{\mathbf{x}}_{0,(i,j)}^{(k-1)}$ is complete.

During training, there are a couple of viable approaches. One method is to stack all bricks and train them end-to-end. Alternatively, a brick-by-brick training methodology can be deployed. In our practical studies, we did not find clear performance differences within these choices.

## D    ADDITIONAL EXPERIMENT RESULTS

### D.1    SKIPPING LEGO BRICKS IN SPATIAL REFINEMENT: COMPARING PG AND PR STRATEGIES

Here we provide the qualitative results to support the analysis of the brick skipping mechanism of LEGO shown in Section 3.2. Progressive growth (PG) and progressive refinement (PR) offer two distinct spatial refinement strategies in the LEGO model, and leveraging their unique properties to skip corresponding LEGO bricks during sampling presents a non-trivial challenge. We visualize the differences between these strategies in Figures 5 and 6. In PG, the model initially uses patch-bricks to generate patches and then employs an image-brick to re-aggregate these local features. Conversely, PR begins by utilizing the image-brick to establish a global structure and then employs local feature-oriented patch-bricks to refine details on top of it. These differences become more evident when $t$ is larger. Based on these observations, we propose that LEGO-PG can skip its top-level image-brick responsible for handling global image information when $t$ is small, while LEGO-PR skips its top-level patch-brick, which is responsible for processing $4 \times 4$ local patch information, when $t$ is large.

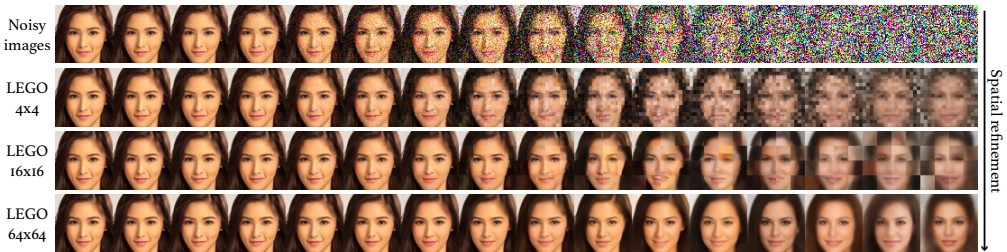

Figure 5: Visualization of the spatial refinement process using the LEGO-PG model: each column represents $\hat{\mathbf{x}}_0^{(k)}$ at different timesteps obtained by corrupting the clean image, shown in the top row of the first column, at various noise levels. Within each column, the spatial refinement process, as described in equation 9, proceeds from the top row to the bottom row.

Below we show analogous results of Figure 4 on different datasets, illustrated in Figures 7-9.

### D.2    ABLATION STUDIES: ON THE EFFECTS OF LEGO BRICKS

In this section, we first study the effects of different LEGO bricks. Note that the patch-bricks target to provide fine-grained details in generation and cost fewer computation resource as they take smaller input resolution and shorter attention span; the image-brick ensures the global structure of the generation and consistency within local regions, but requires more computation cost. Therefore, to

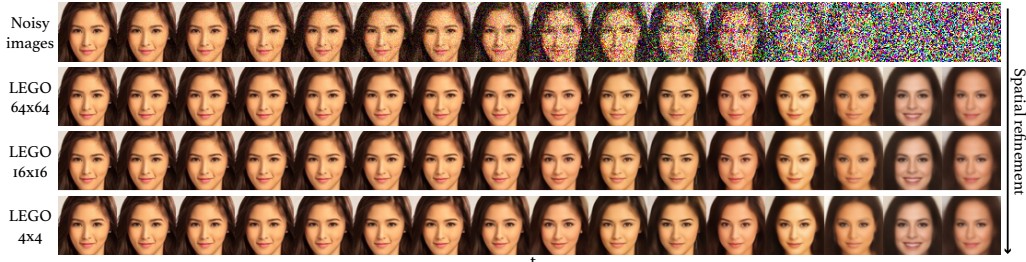

Figure 6: Analogous results of Figure 5, using the LEGO-PR model.

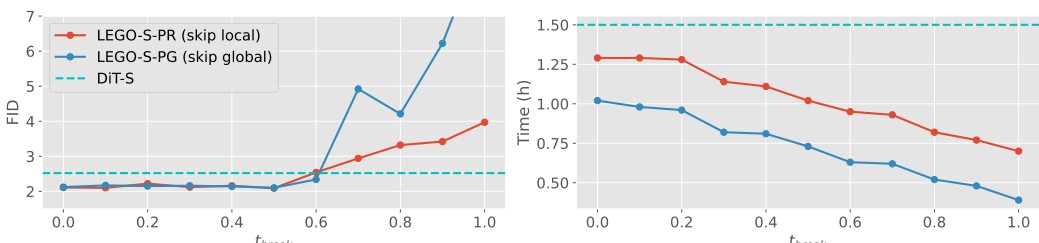

Figure 7: Analogous results to Figure 4. The experiments are conducted using the LEGO-S model with eight NVIDIA A100 GPUs on the CelebA ($64 \times 64$) dataset.

find a good trade-off between generative performance and computation cost, we conduct experiments to find the right balance between the patch-bricks and image-brick.

**Proportion of layers assigned to the image-brick** We first study the trade-off between the training time and the ratio of the depth (number of DiT block layers) of the image-brick to the overall depth of the LEGO model. For simplicity, we consider a LEGO-Diffusion model consisting of a patch-brick with a patch size of $4 \times 4$ and an image-brick with a patch size of $64 \times 64$, both with a receptive-field-size set as $2 \times 2$. We also let both bricks have the same token embedding dimension, number of attention heads, MLP ratio, *etc*. Fixing the total depth as 12, we vary the depth assigned to the image-brick from 0 to 12 and show the results in Figure 10. For a LEGO-Diffusion model consisting of a patch-brick with a patch size of $4 \times 4$ and an image-brick with a patch size of $64 \times 64$, both with a receptive field size set as $2 \times 2$, we visualize how the FID and training time

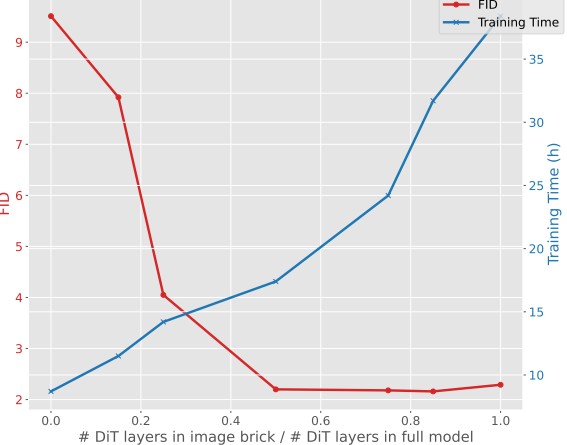

Figure 10: Visualization: how the FID and training time change as the ratio of the depth (number of DiT-based Transformer layers) of the image-brick to that of the patch-brick increases from 0 to 1.

change as the ratio of the depth (number of DiT-based Transformer layers) of the image-brick to that of the patch-brick increases from 0 to 1. Specifically, at the beginning of x-axis, we have a patch-brick with 12 DiT-based Transformer layers and an image-brick with 0 DiT-based Transformer layers, and at the end, we have a patch-brick with 0 DiT-based Transformer layers and an image-brick with 12 DiT-based Transformer layers. We can observe that when we allocate the majority of layers to the patch-brick, the training is efficient but results in a relatively higher FID. As we assign more layers to the image-brick, the FID improves, but the model requires longer training time. Empirically, we find that when the ratio is greater than 50%, the FID improvement becomes less significant, while efficiency decreases at the same time. This observation motivates us to allocate approximately 50% of the layers to the patch-brick.

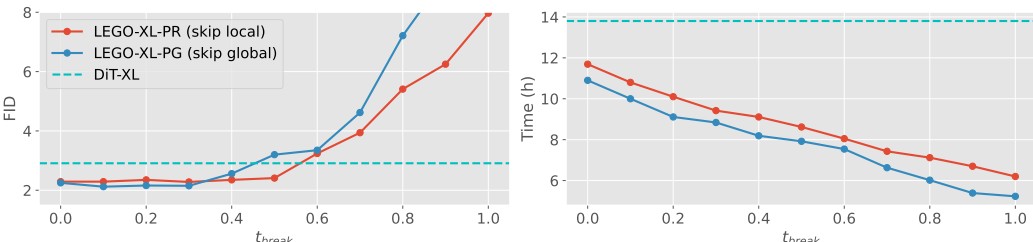

Figure 8: Analogous results to Figure 4, conducted using the LEGO-L model on the ImageNet ($64 \times 64$) dataset with classifier-free guidance.

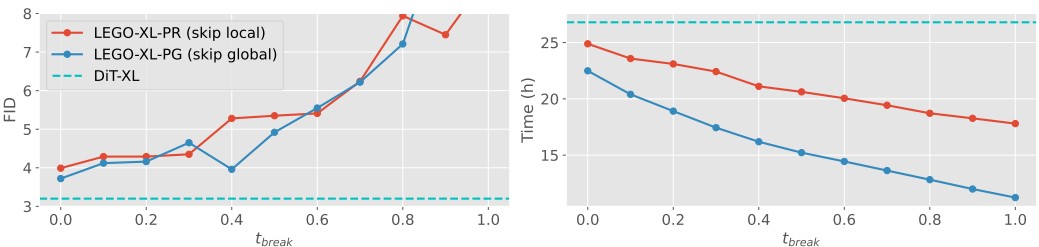

Figure 9: Analogous results to Figure 4, conducted using the LEGO-XL model on the ImageNet ($512 \times 512$) dataset with classifier-free guidance.

**Multi-scale local bricks** Inspired by the multi-stage design commonly found in convolutional networks, we investigate the impact of using multi-scale patch-bricks on our diffusion model's performance. To isolate the effect, we first fix the configurations of both patch- and image-bricks, including the local-receptive-field size that is fixed at $2 \times 2$ and the total number of DiT-block layers is fixed at 12, varying only the resolution of the patches used in training the patch-bricks. Specifically, we experiment with brick sizes of $4 \times 4$, $8 \times 8$, $16 \times 16$, as well as combinations of them. The corresponding FID scores and training times are summarized on the left panel of Figure 11. Our findings indicate that when using a single patch-brick, a $16 \times 16$ brick size yields the best performance but incurs higher training time due to the need for longer token embedding sequences. Using a combination of $4 \times 4$ and $8 \times 8$ patch-bricks does not significantly improve the FID score, possibly because $8 \times 8$ patches do not add substantially more information than $4 \times 4$ patches. However, when a $16 \times 16$ patch-brick is combined with a smaller patch-brick, we see a significant improvement in FID. Notably, the combination of $4 \times 4$ and $16 \times 16$ patch-bricks offers the best trade-off between FID and training time. As a result, this combination of patch sizes is employed when modeling $64 \times 64$ resolutions in our experiments.

To further optimize training efficiency, for the $16 \times 16$ patch-brick, we increase its receptive field size from $2 \times 2$ to $8 \times 8$ and investigate how to assign the six DiT block layers between the $4 \times 4$ patch-brick and the $16 \times 16$ patch-brick. The results shown on the right panel of Figure 11 reveal that assigning more layers to the $4 \times 4$ patch-brick in general results in lower FID but longer training time. Finally, we observe that when the depth of the $16 \times 16$ patch-brick is approximately twice that of the $4 \times 4$ patch-brick, the model achieves a satisfactory balance between performance and efficiency. These observations have been leveraged in the construction of the LEGO model configurations, which are outlined in Tables 5 and 6. It's worth noting that additional tuning of configurations, such as employing specific local receptive field sizes, adjusting the embedding dimension of the local receptive field, varying the number of attention heads, and so on for each LEGO brick, holds the potential for further performance improvements. However, we leave these aspects for future exploration and research.

### D.3 ADDITIONAL GENERATION RESULTS

**Generating beyond training resolution:** Below we first provide additional panorama image generations at resolutions significantly higher than the training images, as shown in Figures 12-14. For all panorama generations (including the top panel in Figure 1), we utilize class-conditional

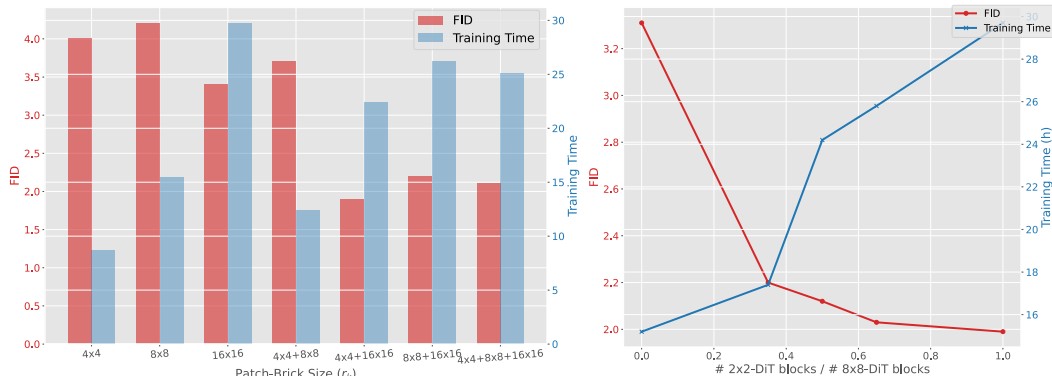

Figure 11: Ablation study on the construction of the patch-bricks when the image-brick is fixed of six layers. (*Left:*) Comparison of three different model configurations: using a single patch-brick with six layers at three different brick sizes, using two patch-bricks, each with three layers, at three different combinations of brick sizes, and using three patch-bricks, each with two layers, each at different patch sizes. (*Right:*) Consider a model comprising two patch-bricks: one with a brick size of $4 \times 4$ and a local receptive field of $2 \times 2$, and the other with a brick size of $16 \times 16$ and a local receptive field of $8 \times 8$. We visualize how the FID and training time change as we vary the ratio of the number of layers assigned to the $4 \times 4$ patch brick to the total number of layers of these two bricks combined, while keeping the total number of layers in the two bricks fixed at six.

LEGO-PG, trained on ImageNet ($256 \times 256$ and $512 \times 512$). As deployed in Bar-Tal et al. (2023), large content is achievable through a pre-trained text-to-image diffusion model trained on images of lower resolution. In our study, we adopt this methodology to show the efficacy of the LEGO framework in such tasks. Notably, the LEGO framework exhibits enhanced capabilities in managing spatial dependencies, which is a critical aspect in this context. For each paranoma generation, we first initialize the latent representation of the panorama image with white Gaussian noise at the desired resolution, such as $(1280/8) \times (256/8) \times 4$ for generating $1280 \times 256$ RGB images in Figure 12. During the generation process, at each timestep, we employ a sliding stride of 7 and have LEGO predict noise within a window of $(256/8) \times (256/8)$ until the target resolution is covered. The predicted noise at each latent voxel is averaged based on the number of times it is generated, taking into account overlaps. When generating images or patterns with a mixture of class conditions, the idea is to produce diverse outputs that adhere to different class characteristics within a single canvas. This could be achieved by dividing the canvas into different regions, where each region corresponds to a specific class condition. In our evaluation, we employed the LPIPS distance metric proposed by Zhang et al. (2018), to assess the similarity between sections of panoramic generations and images from our training set. For each image, we use either a grid-based cropping or a random cropping process. Our analysis involved 100 generated images from various categories. The results are presented in Table 7, including both the average and the standard deviation of the LPIPS distances. Notably, when employing the random cropping method, the quality of panoramas generated using the LEGO model evidently exceeded that of the DiT model. This difference is particularly located in areas comprising outputs from multiple diffusion models, highlighting LEGO's capacity in managing spatial relationships among image patches.

Table 7: Comparision of large-content generation using LPIPS distance (lower is better).

| Crop | Grid - 256 | Random - 256 | Grid - 512 | Random - 512 |
|---|---|---|---|---|
| DiT | $0.15 \pm 0.07$ | $0.66 \pm 0.13$ | $0.55 \pm 0.07$ | $0.76 \pm 0.06$ |
| LEGO-PG | $0.14 \pm 0.07$ | $0.21 \pm 0.17$ | $0.36 \pm 0.07$ | $0.37 \pm 0.06$ |
| LEGO-PR | $0.15 \pm 0.07$ | $0.25 \pm 0.15$ | $0.55 \pm 0.07$ | $0.55 \pm 0.05$ |

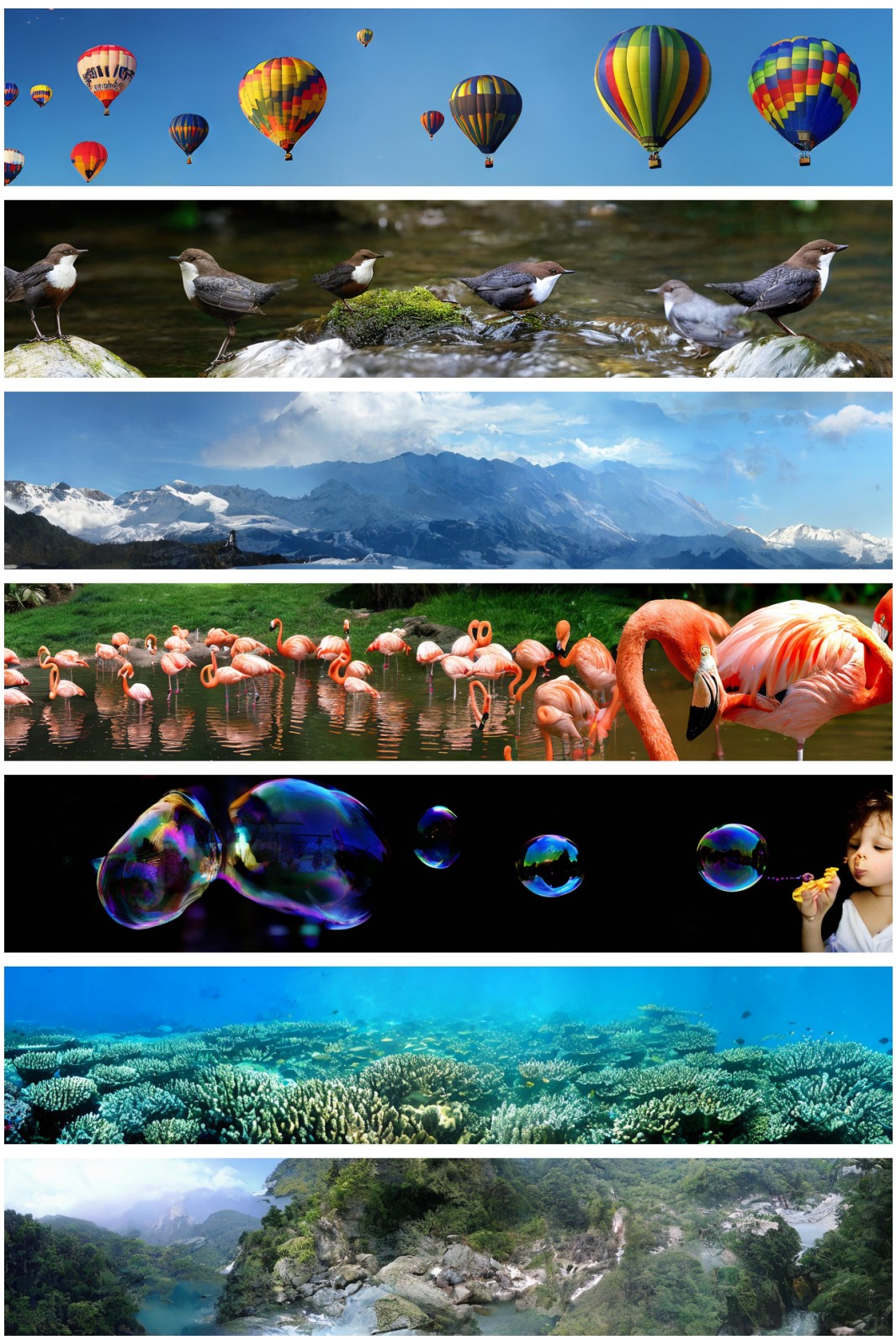

Figure 12: Paranoma generation (1280 × 256) with class-conditional LEGO-PG, trained on ImageNet (256 × 256). (*From top to bottom*) Class: Balloon (index 417), Water ouzel (index 20), Alp (index 970), Flamingo (index 130), Bubble (index 971), Coral reef (index 973), and Valley (index 979).

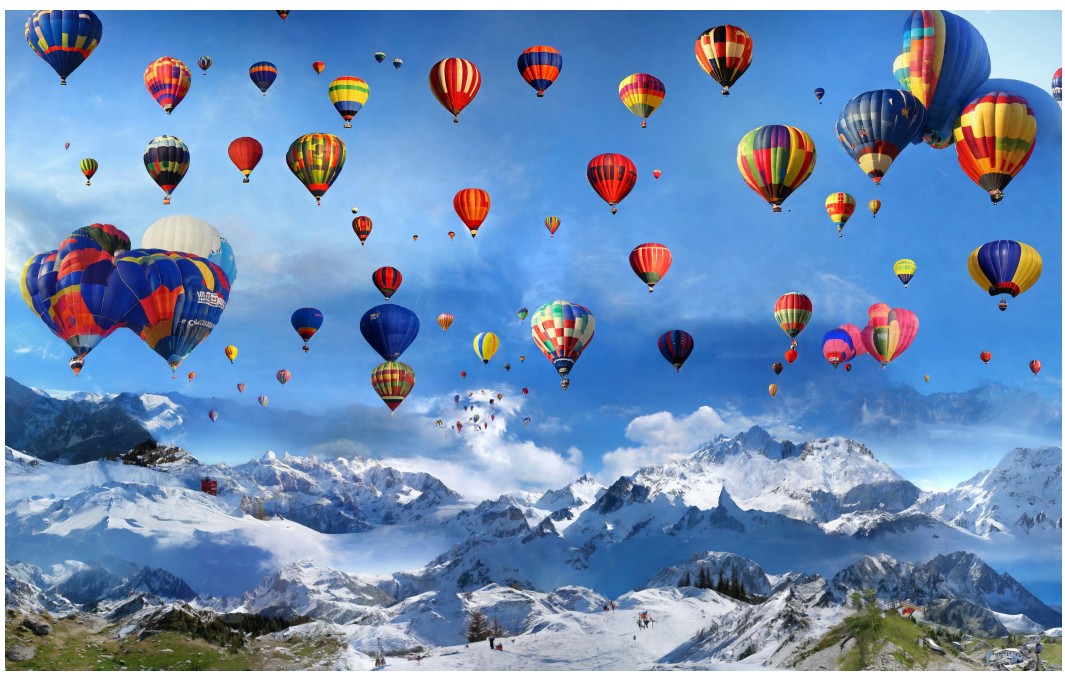

Figure 13: Paranoma generation of 4K ($3840 \times 2160$) resolution with a mixture of class conditions, produced with class-conditional LEGO-PG, trained on ImageNet ($512 \times 512$).

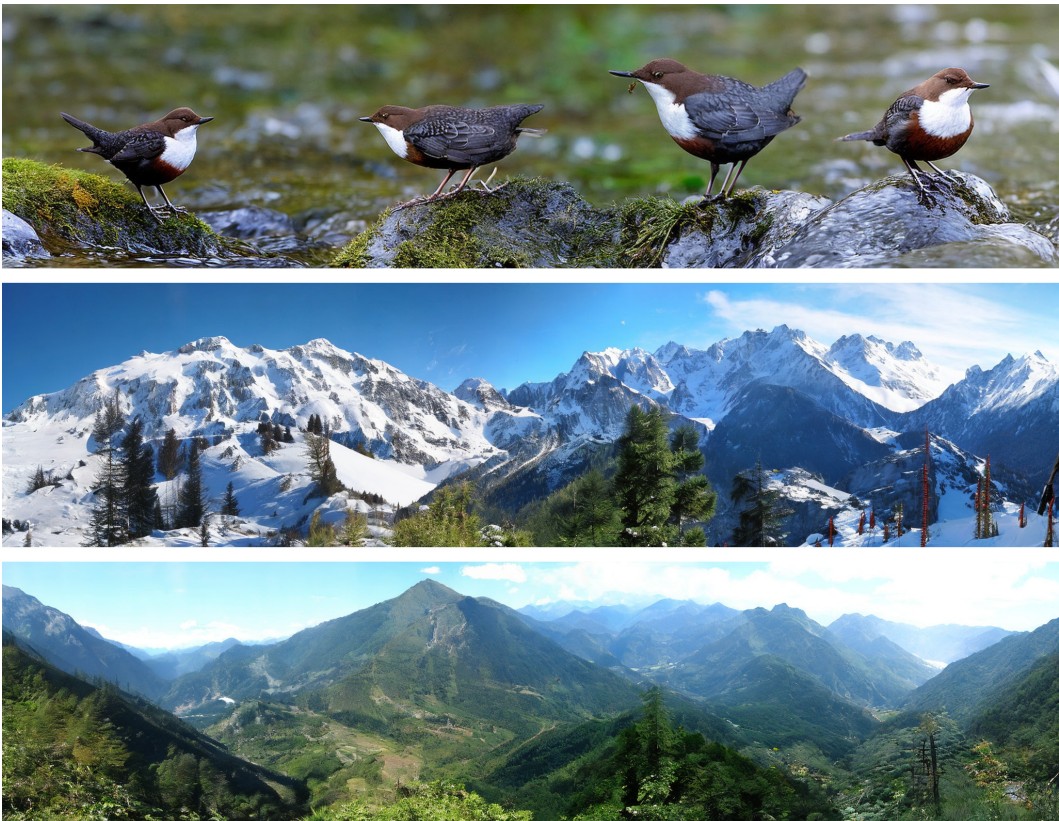

Figure 14: Paranoma generation ($2048 \times 512$) with class-conditional LEGO-PG, trained on ImageNet ($512 \times 512$). (*From top to bottom*) Class: Water ouzel (index 20), Alp (index 970), and Valley (index 979).

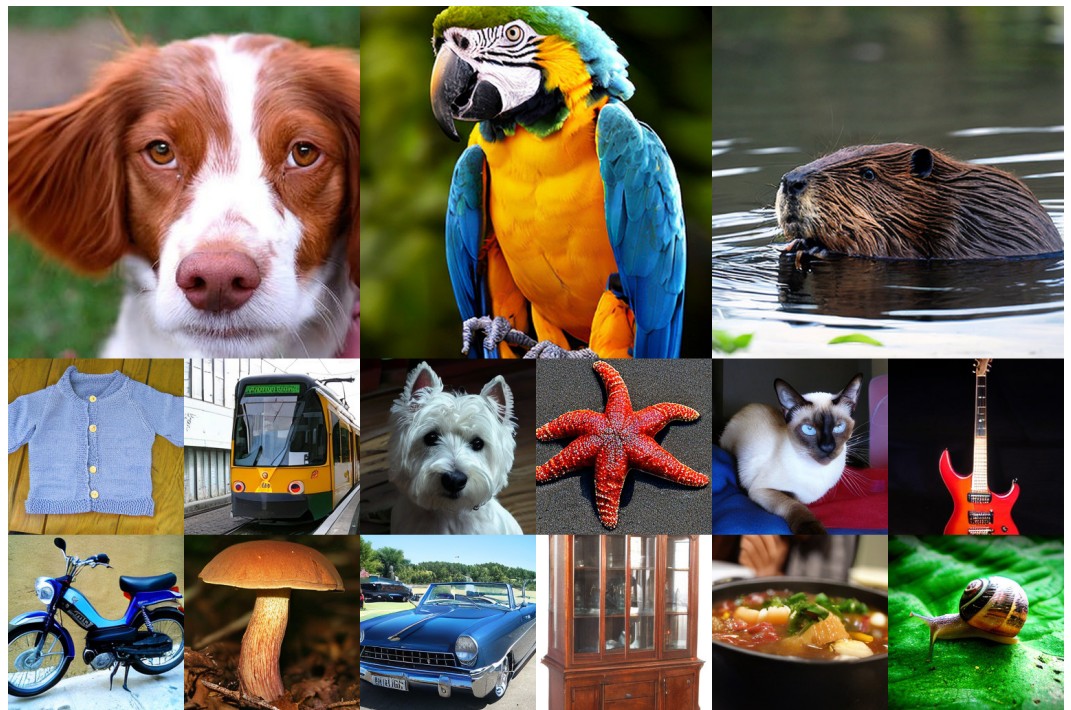

Figure 15: Randomly generated images, using LEGO-PG (cfg-scale=4.0)

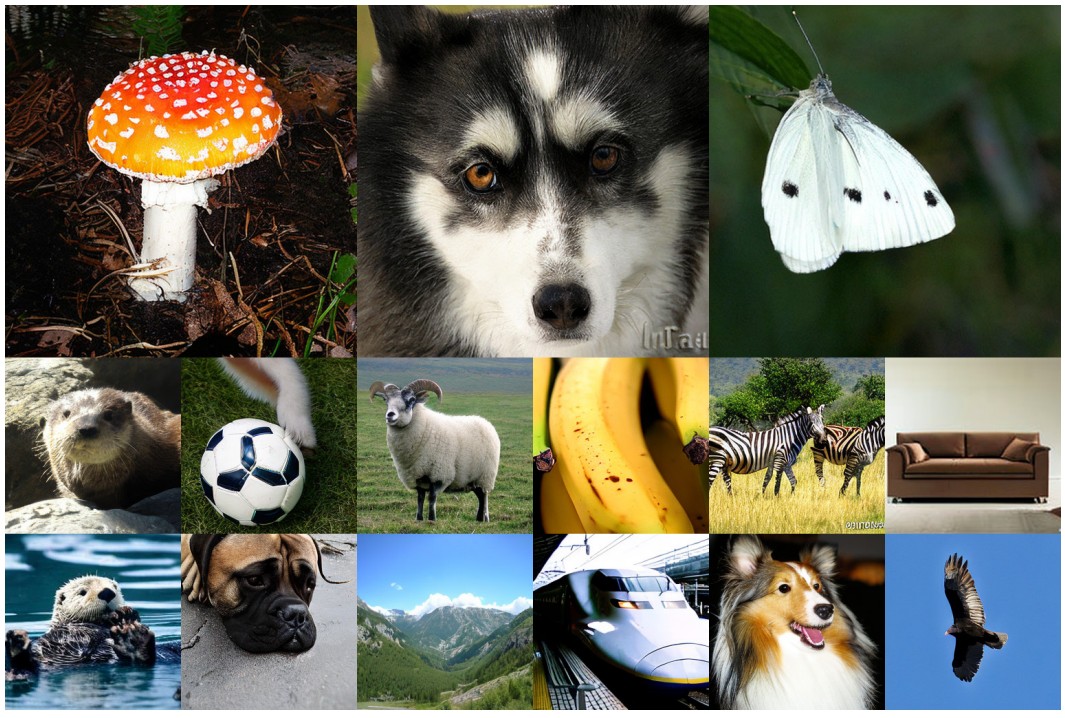

Figure 16: Randomly generated images, using LEGO-PG (cfg-scale=1.5)

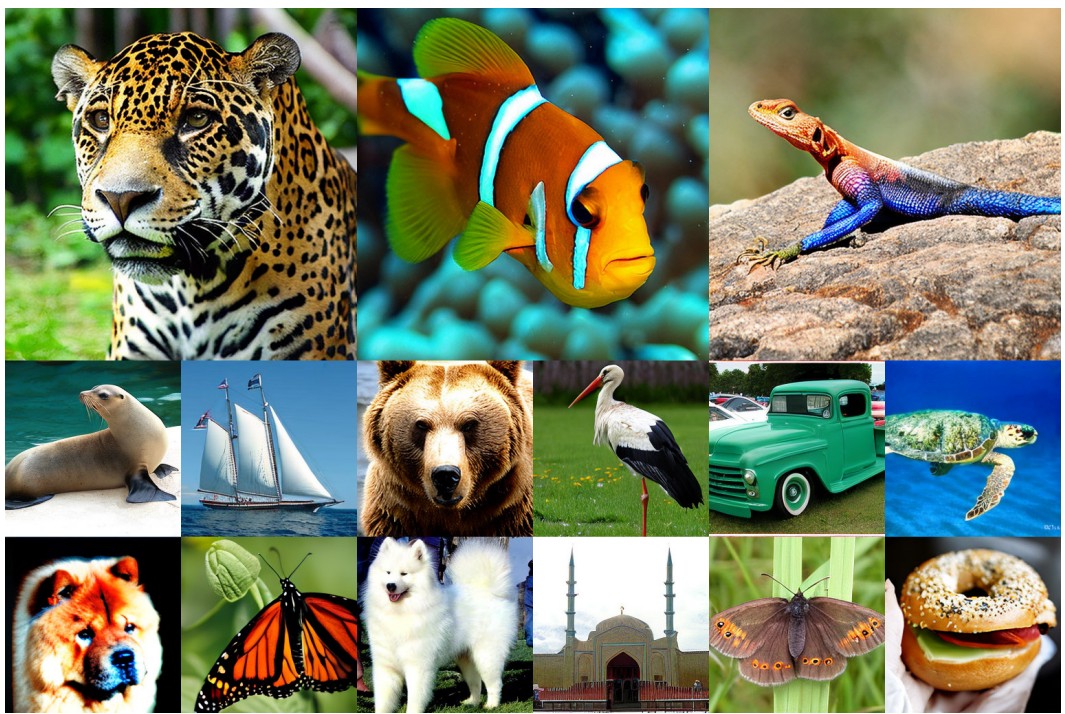

Figure 17: Randomly generated images, using LEGO-PR (cfg-scale=4.0)

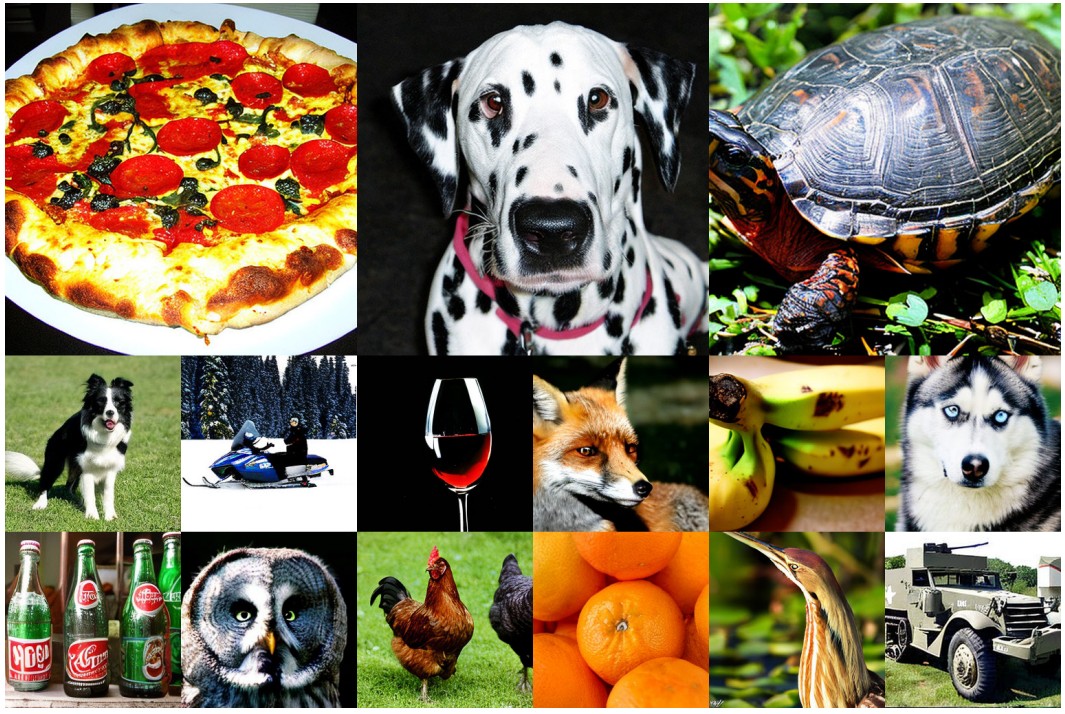

Figure 18: Randomly generated images, using LEGO-PR (cfg-scale=1.5)

