# OpenReview forum: "Learning Stackable and Skippable LEGO Bricks for Efficient, Reconfigurable, and Variable-Resolution Diffusion Modeling"
_ICLR.cc/2024/Conference — ICLR 2024 poster_

### Official Review · Reviewer_ky4P · 2023-10-30

**Soundness:** 3 good
**Presentation:** 3 good
**Contribution:** 3 good
**Rating:** 6
**Confidence:** 3

**Summary:**

This paper proposes LEGO bricks, which can be stacked to create a test-time reconfigurable diffusion backbone, so that at run time one can selectively skip some of the bricks to reduce sampling cost and generate images with higher resolution than the training data.

**Strengths:**

* The idea of designing a run-time configurable backbone for diffusion model is interesting and timely.
* The design within LEGO bricks makes sense, and the performance also looks good.
* The paper is well-written with clear structure. The training and sampling details are presented in a clear way.

**Weaknesses:**

* It seems that the design of LEGO bricks borrows a lot from DiT, so it is a bit unclear to me how much additional contribution w.r.t. network design made in this work.
* It is also clear if the idea of LEGO (skippable and stackable backbone),  is specific to DiT, or it is general enough to be applied to other types of backbone for diffusion models?

**Questions:**

* It is mentioned that the optimization of patch sizes for the LEGO bricks can be future work: can you give some intuition on how to choose the right patch sizes?
* There are two spatial refinement settings mentioned in the paper: PG and PR. So does that mean a model can only take one of these two configurations? or they can be used interchangeably in one model?

---

> ### Author Response · Authors · 2023-11-20
> **Response to Reviewer ky4P (part I)**
>
> Thank you for your positive review and constructive feedback. We're delighted that the concept of LEGO bricks for a reconfigurable diffusion backbone has been well-received. Your appreciation for the paper's clarity and the performance of our design is encouraging. Now, to address your concerns and questions:
>
>
> > Q: What additional network design contributions does LEGO offer compared to DiT?
>
> To better address this question, we summarize the characteristics of Unet, DiT and LEGO in the table below.
>
> | Model                               | Unet        | DiT         | LEGO                      |
> |-------------------------------------|-------------|-------------|---------------------------|
> | Backbone                            | Convolution | Transformer | Convolution / Transformer |
> | Attention                           | Global      | Global      | Local + Global            |
> | Local feature                       | Multi-scale | Fixed-scale | Multi-scale               |
> | Hierarchical structure | Downsample -> Upsample          | No hierachy          |   Patch decomposition (small -> large or large -> small)                   |
> | Multi-variate resolution generation | No          | No          | Yes                       |
>
>
> The primary contribution of our work is the development of the LEGO framework itself. Unlike DiT, which follows a traditional transformer architecture, LEGO introduces a novel, modular approach and is open to deploy both convolutional and transformer basic blocks. This modularity allows for individual components or 'bricks' to be easily skipped or stacked, enabling dynamic architecture reconfiguration based on specific requirements, a feature not inherent in DiT.
>
> In the design of a single LEGO brick, we mainly consider two perspectives: 1) the modular brick needs to be able to handle local-level feature; 2) the local-level processed information needs to be aggregated seamlessly with attention mechanism. Considering these perspectives, DiT blocks well meets these requirements, and we borrow the successful design in DiT, provides a solid base for image processing, to form each LEGO brick.
>
> The entire LEGO model design is however different from DiT, where we can see the multi-scale local feature and local+global attention in the entire model. LEGO optimizes processing power and reduces computational load, particularly in the scenarios where full engagement of all network layers is unnecessary. Moreover, another key aspect of our work is the adaptability of LEGO bricks. LEGO extends this by offering a versatile framework that can be adapted to various types of backbones beyond transformers, such as convolutional networks.
>
>
> > Q: Is the idea of LEGO (skippable and stackable backbone) specific to DiT, or it is general enough to be applied to other types of backbone for diffusion models?
>
> We designed the LEGO framework with versatility in mind, aiming to make it applicable beyond just the DiT blocks used in our main experiments. To demonstrate the general applicability of the LEGO framework, we conducted experiments with the integration of the LEGO concept with a Unet backbone, as shown in Table 4 located in Section 4.3 of the revision.
>
>
> > Q: Some intuition on how to choose the right patch sizes?
>
> We have conducted comprehensive studies in how to choose patch size and the attention length, as detailed in our experimental studies in Appendix C.2. Our core objective was to identify a patch size that maximizes model performance while minimizing computational costs. Through extensive experimentation, we developed a general rule: incrementally increasing the patch size by a factor of four, resulting in a summarized recipe in Tables 4 and 5 of our manuscript.

---

> ### Author Response · Authors · 2023-11-20
> **Response to Reviewer ky4P (part II): LEGO variant, LEGO-U**
>
> > Q: There are two spatial refinement settings mentioned in the paper: PG and PR. So does that mean a model can only take one of these two configurations? or they can be used interchangeably in one model?
>
> Thank you for your insightful question regarding the spatial refinement settings. Indeed PG and PR configurations, while distinct in their approaches, are not mutually exclusive and can indeed be combined within a single model to leverage their respective strengths.
>
> We have been actively developing LEGO-U, a variant that combines the features of LEGO-PR and LEGO-PG. This model processes image patches starting with larger resolutions, transitioning to smaller ones, and then reverting back to larger resolutions. This approach draws inspiration from the architecture of the U-Net, which similarly operates with multiple downsampling and upsampling stages to have multi-scale representations.
>
> We provide some preliminary results of LEGO-U into table below:
>
> | Model             | FID  | sFID          | IS              | Precision | Recall | Iterated Imgs (M) |
> |-------------------|------|---------------|-----------------|-----------|--------|---------------|
> | DiT-XL/2          | 2.27 | 4.60          | 278.24          | 0.83      | 0.57   | 1792          |
> | MDT-XL/2          | 1.79 | 4.57 | 283.01          | 0.81      | 0.6    | 1664          |
> | MaskDiT           | 2.28 | 5.67          | 276.56          | 0.80      | 0.61   | 521           |
> |-------------------|------|---------------|-----------------|-----------|--------|---------------|
> | LEGO-XL-PG (ours) | 2.05 | 4.77          | 289.12          | 0.84      | 0.55   | 512           |
> | LEGO-XL-PR (ours) | 2.35 | 5.21          | 284.73          | 0.83      | 0.60   | 512           |
> | LEGO-XL-U (ours)  | 3.24 | 4.29          | 337.85 | 0.82      | 0.52   | 230           |
>
> Despite the experiments still being underway and not fully completed, these initial findings already demonstrate competitive performance compared to the baselines, LEGO-PR and LEGO-PG. Notably, LEGO-XL-U achieves the best inception score and sFID, indicating the capacity to generate high-quality images.
>
>
> We regard LEGO-U as a distinct project meriting further exploration and study. We welcome your guidance on whether you think it should be included in the current paper.

---

> ### Author Response · Authors · 2023-11-22
>
> Dear Reviewer ky4P,
>
> Thank you for your insightful review and valuable suggestions. We appreciate it if you could review our response by November 22nd, as we will be unable to participate in further discussions after this date. If you have any follow-up questions or require additional information, please feel free to let us know. We greatly appreciate your expertise and the time you've dedicated to our manuscript.
>
> Warm regards,
>
> Authors

---

### Official Review · Reviewer_W6yN · 2023-11-09

**Soundness:** 3 good
**Presentation:** 3 good
**Contribution:** 3 good
**Rating:** 8
**Confidence:** 3

**Summary:**

The paper presents LEGO, Local-feature Enrichment and Global-content Orchestration, an architectural tweak to transformer-based diffusion models for unconditional and class-conditional image generation.
The core idea behind LEGO lies in applying self-attention blocks with varying patch sizes across the transformer, thereby allowing different blocks (called LEGO brick) to focus on different scales of reconstruction / image generation, e.g. blocks with small patches will naturally focus more on local reconstruction, while larger patch sizes encourage enforcing global consistency. Furthermore, each LEGO brick can be applied to a subset of local patches, thereby increasing compute efficiency. Another way in which LEGO allows more efficient generation at test time is to selectively disable some LEGO bricks at specific time steps: For example, the contribution of very local LEGO bricks with small patch sizes is negligible for early t-steps during inference as the local structure of the pixels will still vary heavily in subsequent time steps -- similarly, transformations with a wider receptive field can be dropped at later t-steps since the global structure of the image to be generated has already been decided in earlier time steps. Finally, the method allows variable-size (and aspect ratio) image generation, even after being trained on a fixed-sized image dataset.
Overall, this method achieves competitive results with favourable compute cost.

**Strengths:**

- The proposed method appears to be original and is intuitive.

- The authors present sensible experiments that ablate over several design choices of their method. Especially the choice of dropping specific LEGO blocks during inference is quite interesting.

- The authors clearly state several limitations of their work, none of which are a major concern for this submission -- the paper is well-scoped.

- The writing style is good, and the authors always try to simplify and add intuition to design choices.

**Weaknesses:**

- The method presentation could be improved. Intro, Section 3.1 and Figure 3 provide some high-level intuition which is helpful for the start. Meanwhile the remaining sections obfuscate major questions, e.g. whether a LEGO brick is applied densely or sparsely or how the patches are selected.

- The majority of the experiments and results are placed in the appendix, while a very large portion of the main paper is dedicated on an extensive introduction, related works, and more context setting at the start of the method section. The authors should strongly consider making the first 4 pages of the main paper significantly more concise, thereby allowing the main findings to move from the appdix to the experimental section of the main paper.

- The graphic shown in right panel in Figure 1 is interesting, but there may be a better choice of representing the same data. Drawing circles with FLOPs as their radius makes differences between models appear much more significant than they are (since the difference in circle size == circle area grows with the square of the radius), making the plot somewhat misleading. A more common choice is a plot showing FID on the y-axis and FLOPs on the x-axis, where each model would be a single dot.

**Questions:**

- How many patches are selected (for a single LEGO brick)? During training time & inference.

- Page 1: "This requirement arises from the model's need to learn how to predict the mean of clean images conditioned on noisy inputs [...]". Could the authors provide a citation for this claim?

---

> ### Author Response · Authors · 2023-11-20
> **Response to Reviewer W6yN**
>
> Thank you for your detailed and constructive review. We are grateful for your acknowledgment of LEGO's originality and your suggestions in polishing the presentation. We have made revision accordingly, and provide our response to your concerns point-to-point.
>
> > Q: The method presentation could be improved. The majority of the experiments and results are placed in the appendix.
>
> We have re-organized the presentation of our paper, making more space to the experiments. Specifically, we move some high-level review and discussions to Appendix A. We move the study of the sampling with skippable LEGO bricks and the study in terms of the versatility of LEGO in section 4.2 and 4.3.
>
> > Q: The graphic shown in right panel in Figure 1 can be improved.
>
> Following your suggestion, we update the right figure of Figure 1 (Figure 2 in the revised paper). The updated figure now features a plot with FLOPs on the x-axis (presented with log scale).
>
>
> > Q: How many patches are selected (for a single LEGO brick)? During training time & inference.
>
> In our main experiments, 50% and 75% of all non-overlapping patches are used in the training of patch-bricks on CelebA and ImageNet datasets. In the inference time, as we have no patches from real images that can be leveraged, we need to use all patches for the generation, but we can skip bricks to save computation cost.
>
> > Q: Reference for: Page 1 - "This requirement arises from the model's need to learn how to predict the mean of clean images conditioned on noisy inputs [...]".
>
> We have clarified that the concept of “diffusion models learn how to predict the mean of clean images conditioned on noisy inputs” can be derived from two theoretical foundations: either Tweedie’s formula (Robbins, 1992; Efron, 2011), as illustrated in Luo, 2022 and Chung et al., 2022, or the Bregman divergence (Banerjee et al., 2005), as revealed in Zhou et al. (2023).

---

> > ### Comment · Reviewer_W6yN · 2023-11-20
> > **Thanks for the reply**
> >
> > I thank the authors for the reply and for incorporating the feedback. The new structure, putting a higher emphasis on the experiments in the main paper, is a welcome change.
> > While the computational gain by dropping certain parts of the model at inference time is not extreme, I believe the paper can be valuable to the community and thus merits publication; pending final comments by the other reviewers.

---

> > > ### Author Response · Authors · 2023-11-20
> > >
> > > We are sincerely grateful for your recognition of the improvements we've implemented in our manuscript based on your feedback, and for your decision to raise the evaluation score.
> > >
> > > LEGO introduces several new features, with test time reconfigurability (adjusting model
> > > sizes without the need for retraining/finetuning) being one of them.  To demonstrate this, we reconfigured the model to optimally balances sampling speed with performance.  Envisioning real-world applications, we are actively studying the potentials of a LEGO-based reconfigurable system deployed in the cloud, which is designed to adapt in real-time to various user needs. It will offer flexibility in real-time model resizing, accommodating a wide range of users with diverse budgetary and quality needs.
> > >
> > > We are eagerly anticipating additional feedback from the other reviewers and remain dedicated to continuously refining our work in accordance with their suggestions.

---

### Official Review · Reviewer_x9WT · 2023-11-09

**Soundness:** 3 good
**Presentation:** 3 good
**Contribution:** 2 fair
**Rating:** 6
**Confidence:** 3

**Summary:**

The paper is concerned with diffusion-based image generation and proposes Local-feature Enrichment and Global-content Orchestration (LEGO) blocks. These blocks can be flexibly arranged to process local patches of different sizes, thereby implementing a hierarchical structure. The authors also envision skipping and recombining these blocks at training and inference time, as well as incorporating pretrained diffusion models into the structure of blocks. The proposed approach is evaluated on Celeb-A face generation and ImageNet class-conditional generation and compared with popular methods from the literature.

**Strengths:**

The paper is well written and generally easy to follow. To my knowledge, the idea of splitting the image into patches and to process with a hierarchy of modules has not been explored in the diffusion literature before. The skipping and mixing of modules envisioned by the authors is interesting. The method seems to train more efficiently than recent diffusion methods and has lower inference FLOPs than those at the same sample quality.

**Weaknesses:**

It seems the authors in essence propose a block-based hierarchical architecture which is not very different from a UNet. While there is lots of talk on modularity and skipping of blocks, these aspects are only explored in the appendix on 64x64 CelebA images, i.e. an easy data set at a resolution where inference speed ups are not very interesting. The aspect of incorporating a pretrained diffusion model is only explored as an ablation. Further, generating images larger than the training resolution is demonstrated with a few examples, which might also be obtained by cleverly leveraging prior works.

Given all these points it feels like the contribution is not as significant as suggested at the beginning of the paper. Experiments where skipping blocks at inference time yield substantial wallclock-time speedups for class conditional ImageNet generation at 256 or 512 pixels resolution would be more convincing.

Minor: Page 6 typo: LEBO

**Questions:**

- How is the FID computed? FID numbers can differ substantially depending on the implementation, and whether the train or validation set is used as a reference. For example DiT and ADM both use the ADM Tensorflow evaluation suite.
- What is a “linear MLP” (page 7)? How does it differ from a linear layer?
- How do the parameter counts of the proposed model compare with the baselines for the ImageNet 256 and 512 pixels experiments? I could only find parameter counts for the 64 pixel models.
- Are the bricks of a given stage shared across space, or are they specialized per patch?

---

> ### Author Response · Authors · 2023-11-20
> **Response to Reviewer x9WT (part I)**
>
> Thank you for your valuable feedback. In what follows, we provide detailed responses to each of your concerns.
>
> > Q: Similarity to UNet? The proposed block-based hierarchical architecture was viewed as not being significantly different from a UNet structure.
>
> We summarize the characteristics of Unet, DiT and LEGO in the table below.
>
> | Model                               | Unet        | DiT         | LEGO                      |
> |-------------------------------------|-------------|-------------|---------------------------|
> | Backbone                            | Convolution | Transformer | Convolution / Transformer |
> | Attention                           | Global      | Global      | Local + Global            |
> | Local feature                       | Multi-scale | Fixed-scale | Multi-scale               |
> | Hierarchical structure | Downsample -> Upsample          | No hierachy          |   Patch decomposition (small -> large or large -> small)                   |
> | Variable resolution generation | No          | No          | Yes                       |
>
> LEGO distinguishes itself from Unet in several key aspects:
>
> - The LEGO bricks that constitute the LEGO network are distinct from the convolutional filters employed in UNet. Unlike UNet, which relies heavily on convolutional layers, the LEGO model is flexible to adopt either convolutional or transformer blocks.
>
> - The self-attention deployed in the Unet is global-level, while LEGO adopts a different approach, which makes its self-attention length vary depending on the 'LEGO brick'. In some cases, attention is confined to specific image patches defined by the brick size, while in some other bricks, it extends over the entire image. This selective mechanism allows for more focused and efficient feature aggregation.
>
> - Both models extract multiscale features, but they achieve this goal differently. The Unet uses a series of downsampling and upsampling stages, whereas LEGO leverages varied patch decomposition methods. This distinction in processing strategy results in diverse feature representations.
>
> - A notable difference is that each 'brick' in LEGO can be trained to independently generate patch-wise outputs, resulting in variable resolution generation, a capability not inherently present in Unet's intermediate features. This property of LEGO bricks enhances their utility in diverse generative tasks.
>
> - The variants LEGO-PG and LEGO-PR, as discussed in this paper, do not arrange the LEGO bricks in a manner akin to Unet. The LEGO-U variant, mentioned in the 'Overview of Revisions,' draws inspiration from Unet in its organization of local units. However, even in LEGO-U, the construction of these local units remains distinct from those in Unet.
>
> > Q: Limited Exploration of Modularity and Skipping Blocks? “The modularity and skipping of blocks are only explored in the appendix on 64x64 CelebA images, which is an easy data where inference speed ups are not very interesting.  Experiments where skipping blocks at inference time yield substantial wallclock-time speedups for class conditional ImageNet generation at 256 or 512 pixels resolution would be more convincing.”
>
> We move the figure presenting the on the ImageNet 256x256 data to Figure 4 (Figure 7 of the original submission). These experiments demonstrate similar effects shown in the experiments on CelebA, but also validate the effectiveness of our approach in a more complex and high-resolution setting.
>
> To further substantiate our findings, we have now included additional analyses on ImageNet 64x64 and ImageNet 512x512 in Figure 8-9. These are similar in spirit to Figures 4. These results provide more comprehensive evidence of the efficiency gains achievable through our model, especially in terms of wallclock-time speedups during class-conditional ImageNet generation at resolutions of 64, 256 and 512 pixels.

---

> ### Author Response · Authors · 2023-11-20
> **Response to Reviewer x9WT (part II)**
>
> > Q: “Generating images larger than the training resolution is demonstrated with a few examples, which might also be obtained by cleverly leveraging prior works.”
>
> We appreciate your suggestion to consider prior works in the context of generating images larger than the training resolution. Following your advice, we have now included a reference to the MultiDiffusion work in our manuscript. This work deploys techniques similar to those we described in Appendix C.4, particularly in generating large-content images using text-to-image diffusion models.
>
> Since our case is to generate large-content with class-conditional model pre-trained with smaller resolution, we compare the large-content generation using pre-trained DiT model on ImageNet 256 x 256 and 512 x 512. To provide a thorough comparison, we generated images using our method and then compared the LPIPS distance (a perceptual score) with those of local crops:
>
>
> | Crop    | Grid - 256   | Random - 256 | Grid - 512   | Random - 512 |
> |---------|--------------|--------------|--------------|--------------|
> | DiT     | 0.15 &plusmn; 0.07 | 0.66 &plusmn; 0.13 | 0.55 &plusmn; 0.07 | 0.76 &plusmn; 0.06 |
> | LEGO-PG | 0.14 &plusmn; 0.07 | 0.21 &plusmn; 0.17 | 0.36 &plusmn; 0.07 | 0.37 &plusmn; 0.06 |
> | LEGO-PR | 0.15 &plusmn; 0.07 | 0.25 &plusmn; 0.15 | 0.55 &plusmn; 0.07 | 0.55 &plusmn; 0.05 |
>
>
>
>
> The results of this comparison, which we have included in the revised manuscript, demonstrate that our method is effective in generating high-quality, large-resolution images by orchestrating the generated small patches.
>
>
> > Q: How is the FID computed?
>
> We use the same evaluation suite, i.e., the ADM Tensorflow evaluation suite and their reference batch, ensuring that our evaluations are rigorous and reliable. We have added these details in Appendix D.
>
> > Q: What is a “linear MLP”?
>
> Thank you for highlighting the ambiguity in our use of the term "linear MLP." To avoid confusion and ensure clarity in our manuscript, we have revised the text to specifically mention "a linear layer" instead of "linear MLP."
>
>
> > Q: How do the parameter counts of the proposed model compare with the baselines for the ImageNet 256 and 512 pixels experiments?
>
> The parameter counts are listed below, we have included them in the Tables 5-6 of the revised manuscript:
>
> | Model   | Total Param  | Param with skipping (PG) | Param with skipping (PR) |
> |---------|--------------|--------------------------|--------------------------|
> | LEGO-S  | 35           | 16                       | 26                       |
> | LEGO-L  | 464          | 223                      | 331                      |
> | LEGO-XL | 681          | 327                      | 555                      |
>
>
> > Q: Are the bricks of a given stage shared across space, or are they specialized per patch?
>
> In the current design of our model, each brick at a specific stage is indeed shared across the entire spatial domain. This means that the same brick processes all possible patches within that stage.

---

> ### Author Response · Authors · 2023-11-22
>
> Dear Reviewer x9WT,
>
> As we will not be able to provide additional results or modifications to our manuscript after November 22nd, we are reaching out to see if our response adequately addresses your concerns.
>
> We greatly value your comments and are committed to refining our work. Any further feedback would be helpful in polishing our final revisions.
>
> Thank you for your time and expertise.
>
> Sincerely,
>
> Authors

---

> ### Comment · Reviewer_x9WT · 2023-11-22
>
> Thanks to the authors for their detailed rebuttal.
>
> I now have fewer concerns about the benefits of skipping blocks given the quantitative results on ImageNet highlighted in the rebuttal. Overall, I'm leaning towards acceptance.
>
> I appreciate that the authors report the training wall clock time as a function of t_break. The only remaining evaluation that I think is really important for the final version of the paper is inference time (actual runtime on accelerator, not FLOPs) vs sample quality/FID, for different t_break.

---

> > ### Author Response · Authors · 2023-11-22
> >
> > Thank you for your follow-up comments and for increasing the score.
> >
> > We would like to clarify that the right panels of Figures 4, 7, 8, and 9 are intended to illustrate the inference time required for generating 50,000 images, with varying values of $t_break$. We have updated the captions of these figures to convey this information more explicitly.
> >
> > Please inform us if there has been any misinterpretation of your comments or if further modifications are needed for the final version of our paper.

---

### Author Response · Authors · 2023-11-20
**Overview of Revisions**

Dear Reviewers,

Thank you for your valuable feedback. In response, we have carefully revised our paper, with all major changes highlighted in blue. This revision not only enhances the paper's organization but also introduces new findings. In particular, we have shifted several key results into the main body of the paper. In terms of generating higher-resolution images, we devised a method for a quantitative comparison between LEGO and DiT and provide additional generations results with resolution ranging from 2048x512 to 4k.  Additionally, in our response to Reviewer ky4P, we have detailed some encouraging initial results of a novel LEGO variant, LEGO-U, which utilizes a Unet-like structure for its LEGO bricks. We are open to including LEGO-U in the main paper, should you consider it necessary.

---

> ### Comment · Area_Chair_5KqV · 2023-11-20
> **Rebuttal**
>
> Reviewers, PTAL at the rebuttal.

---

### Meta-Review · Area_Chair_5KqV · 2023-12-05

**Metareview:**

The authors aim to improve diffusion-based image generation and propose Local-feature Enrichment and Global-content Orchestration (LEGO) blocks. These blocks can be flexibly arranged to process local patches of different sizes, thereby implementing a hierarchical structure. This approach allows skipping and recombining these blocks at training and inference time, as well as incorporating pretrained diffusion models into the structure of blocks. The proposed approach is evaluated on ImageNet class-conditional generation and the Celeb-A dataset, and improvements in FLOPs and wallclock time for different configurations is reported.

The reviewers found the idea novel and relevant to the image generation audience at ICLR and provided several missing ablations in the rebuttal and during the discussion phase. Given the updated state of the manuscript, I will recommend acceptance.

**Justification For Why Not Higher Score:**

I would like to see more evidence for dramatic wall-clock time wins.

**Justification For Why Not Lower Score:**

Novel idea, good potential building block (no pun intended) for future models, at least at inference time.

---

### Decision · Program_Chairs · 2024-01-16

Accept (poster)